# Optical characteristics after Descemet membrane endothelial keratoplasty: 1-year results

**Takahiko Hayashi** [1,2,3,4,5] *, **Akira Kobayashi** [6], **Hidenori Takahashi** [3], **Itaru Oyakawa** [7], **Naoko Kato** [1,8], **Takefumi Yamaguchi** [9]

**1** Department of Ophthalmology, Yokohama Minami Kyosai Hospital, Yokohama, Kanagawa, Japan, **2** Department of Ophthalmology, University of Cologne, Cologne, Germany, **3** Department of Ophthalmology, Jichi Medical University, Shimotsuke, Tochigi, Japan, **4** Department of Ophthalmology, Yokohama City University School of Medicine, Kanagawa, Japan, **5** Department of Technology and Design Thinking for Medicine (DT2M), Hiroshima University, Higashihiroshima, Japan, **6** Department of Ophthalmology, Graduate School of Medical Science, Kanazawa University, Kanazawa, Japan, **7** Department of Ophthalmology, Heart Life Hospital, Nakagami-gun, Okinawa, Japan, **8** Minamiaoyama Eye Clinic, Minato-ku, Tokyo, Japan, **9** Department of Ophthalmology, Tokyo Dental College, Ichikawa General Hospital, Chiba, Japan

\* takamed@gmail.com

**Data Availability Statement:** All relevant data are within the manuscript and its Supporting Information files.

## Abstract

### Purpose

To evaluate the corneal characteristics after Descemet membrane endothelial keratoplasty (DMEK) compared with normal corneas.

### Methods

Patients who underwent DMEK at Yokohama Minami Kyosai Hospital were included and prospectively evaluated pre-operatively and at postoperative months 1, 3, 6, and 12, and compared to healthy controls. Corneal characteristics evaluated included corneal curvature (keratometric value [KV]; D), central corneal thickness (CCT), peripheral corneal thickness (PCT), and corneal higher-order aberrations [HOAs] at 6.0 mm diameter, calculated by anterior segment optical coherence tomography and logarithm of the minimal angle of resolution [logMAR].

### Results

A total of 30 eyes of 30 patients (6 men, 24 women, mean age 73.4 ± 7.4 years) were included and compared with 31 age-matched healthy control eyes (13 men, 18 women; mean age 73.0 ± 6.7 years). LogMAR after DMEK improved from 0.87 ± 0.07 preoperatively to 0.04 ± 0.07 at 12 months postoperatively (p<0.001). Although anterior KVs of DMEK eyes were similar to those of control eyes, posterior KVs were significantly larger (-6.4 ± 0.3 D vs. -6.3 ± 0.2 D; p = 0.02). Total HOAs after DMEK improved from 1.94 ± 1.05 μm preoperatively to 1.05 ± 0.16 μm at 12 months postoperatively (p<0.001), which was significantly higher than that in control eyes (0.63 ± 0.06) (p<0.001). Despite the similar CCTs in the two groups, the PCT was significantly larger in DMEK eyes (704 ± 41 μm vs 669 ± 38 μm, p = 0.002) at 12 months.

**Funding:** The funders had no role in study design, data collection and analysis, decision to publish, or preparation of the manuscript. This work was supported by Alexander von Humboldt Foundation (https://www.humboldt-foundation.de/web/home.html) and the Japan Eye Bank Association (http://www.j-eyebank.or.jp), but did not have any additional role in the study design, data collection and analysis, decision to publish, or preparation of the manuscript.

**Competing interests:** The authors have declared that no competing interests exist.

## Conclusion

Despite achieving good visual function and excellent corneal clarity, eyes that underwent DMEK showed a steeper posterior KV and higher corneal HOAs than normal eyes even at 12 months after surgery.

## Introduction

Corneal transplantation has evolved significantly since the first full-thickness keratoplasty was performed by Dr. Eduard Zirm in 1905 [1]. However, this technique is associated with disadvantages such as graft rejection, glaucoma (steroid-dependent) and suture-related problems, slow and low visual recovery with significant astigmatism, infection, or risk of globe rupture with trauma [2]. For patients with endothelial dysfunction, a partial-thickness endothelial keratoplasty has now become standard of care [3–5].

In particular, Descemet's membrane endothelial keratoplasty (DMEK), developed by Dr. Gerrit Melles, has resulted in the best visual outcomes for these patients [6]. In this procedure, the diseased endothelium and Descemet's membrane are replaced in an anatomically precise fashion. DMEK offers two primary advantages: 1) rapid visual recovery with a better final visual outcome than other keratoplasty techniques [7, 8], and 2) an extremely low incidence of graft rejection even when compared to Descemet's stripping automated endothelial keratoplasty (DSAEK) [9–11]. Previous studies have proved the superiority of DMEK to either DSAEK or ultra-thin DSAEK in terms of visual function [12–16].

However, some patients show variations in visual outcomes despite presenting with a completely clear cornea. Previous reports have described factors associated with visual function and the specific changes in the posterior cornea after DMEK [12–16], with corneal backscatter and higher-order aberrations (HOAs) as main factors [12–20]. In addition, better preoperative visual acuity is correlated with better postoperative visual acuity [7]. We have already reported that topographic changes and improvements in irregularity after keratoplasty or corneal disease show a strong correlation with recovery of visual function [15, 17–23].

Herein, we investigate the time course alterations in corneal shape after DMEK in comparison with normal healthy corneas. To the best of our knowledge, this is the first report showing the time course of the development of HOAs and status of corneal curvature one year after DMEK in comparison with normal control eyes.

## Materials and methods

### Study design

This prospective study was approved by the Yokohama Minami Kyosai Review Board (approval no. YKH_29_03_05) and was performed in accordance with the ethical standards as laid down in the 1964 Declaration of Helsinki and its later amendments. The study procedures followed all institutional guidelines, and all patients provided informed consent in writing. Patients indicated for DMEK from July 2017 to Mar 2018 were enrolled. Our inclusion criteria were patients who had pseudophakic eyes without stromal scarring but with stromal edema. Phakic DMEK or triple DMEK (combined with cataract surgery) were excluded because of the difference in procedure. Patients with visual limitations such as amblyopia, glaucoma, or macular disease were excluded, as were cases in which DMEK was performed for failed PKP, because of the presence of an irregular corneal surface or greater astigmatism. Age-matched phakic eyes without history of ocular surgery or ocular disease were selected as healthy controls.

## Surgical technique

All patients underwent DMEK surgery by a single surgeon (T.H.), as previously described [24–26]. Briefly, after the creation of a descemetorhexis with an approximate diameter of 9.0-mm under air, an appropriately under-sized graft (7.75 mm, 8.0 mm, or 8.25 mm) was inserted using an intraocular lens inserter (WJ-60M®; Santen, Osaka, Japan), and was unfolded and fixated with 20% sulfur hexafluoride ($SF_6$) until 80% of the anterior chamber volume was filled. Pre-stripped donor corneas obtained from Cornea Gen (https://corneagen. com/) were used in this study. In eyes with epithelial disorders such as corneal erosion or bullae before surgery, epithelial debridement was performed. All patients strictly attended the follow-up visits as per standard protocols. Rebubbling was performed in cases with large and progressive graft detachments.

## Patients and examinations

In addition to standard examinations performed using slit-lamp microscopy, the following factors were evaluated preoperatively and at 1, 3, 6, and 12 months postoperatively in all eyes: 1) topographic factors determined by anterior segment optical coherence tomography (AS-OCT; CASIA, Tomey, Nagoya, Japan); aberration factors (HOAs, spherical aberrations [SAs], and coma aberrations [Comas] at a 6.0-mm diameter in the anterior-, posterior-, and total cornea); keratometric values (KV of the anterior, posterior, and total cornea; diopter [D]); corneal thickness (central corneal thickness [CCT] and peripheral corneal thickness [PCT] at 9.0 mm); and best spectacle-corrected visual acuity (BCVA; logarithm of the minimal angle of resolution [logMAR]). Outcomes were compared with healthy controls.

## Statistical analyses

Statistical analyses were performed using JMP Pro software version 14.0.0 (SAS Institute, Cary, NC, USA). For statistical analysis, BCVA was converted to logMAR units. For comparing the continuous variables in each group (patient age, BCVA, CCT, PCT, HOAs, Comas, SAs), we used either a one-way analysis of variance (ANOVA) or Mann–Whitney U test, while nominal variables such as patient sex and operated eye were compared using Pearson's chi-square test. The Wilcoxon test was used to compare the mean values preoperatively and postoperatively, where appropriate. Correlations between BCVA and all aberration factors (HOAs, Comas, and SAs in anterior, posterior, and total cornea) were evaluated using Spearman's correlation test. Statistical analyses were performed using G*Power 3 software (http://www. gpower.hhu.de/en.html), with a statistical power (1-β) of 0.95 and an α level of 5% (based on a two-sided Wilcoxon–Mann–Whitney test indicating the difference between two independent means prior to the experiments). Statistical significance was defined as $p < 0.05$. All average values are presented as the mean ± standard deviation (SD).

## Results

### Sample size

From our calculations using unpublished preliminary data—the HOAs of DMEK eyes at one year and of control eyes were 0.97 ± 0.43 μm and 0.61 ± 0.19, respectively—the effect size was 1.082. Using this value with an α error probability of 0.05 and power (β) of 0.95, the estimated sample size was 25. Therefore, we set the sample size as 30, and 30 patients were followed up with during the study period (from July 2017 to March 2018). Thirty-one phakic controls were obtained and compared with DMEK patients during the same period.

## Patient characteristics

Table 1 summarizes the characteristics of the 30 DMEK patients (24 women and 6 men; 73.4 ± 7.4 years old) and the 31 controls (18 women and 13 men; 73.0 ± 6.7 years old) included in the study. There were no statistically significant differences in age (p = 0.829) or sex between the two groups (p = 0.097). All patients were of Japanese ethnicity. The indications for DMEK were mainly pseudophakic bullous keratopathy (PBK, n = 21) and Fuchs endothelial corneal dystrophy (FECD, n = 9), and. Rebubbling was performed in two eyes (6.7%) within 2 weeks after surgery, after which complete attachment was achieved. Preoperatively, all eyes showed epithelial disorders, such as bullae or microcystic changes due to endothelial dysfunction, which resolved after DMEK.

## Visual acuity and corneal thickness

BCVA significantly improved after DMEK throughout the postoperative period as follows: from 0.87 ± 0.52 preoperatively to 0.26 ± 0.23 at 1 month (Table 2, p < 0.001), 0.12 ± 0.15 at 3 months (p < 0.001), 0.05 ± 0.11 at 6 months (p < 0.001), and 0.04 ± 0.11 (p < 0.001) at 12 months postoperatively. However, BCVA was still inferior to that of normal controls at all time points.

CCT after DMEK significantly decreased throughout the postoperative period up to 6 months after surgery. At 12 months after surgery, it was comparable to normal controls. CCT significantly decreased from 682 ± 100 μm preoperatively to 523 ± 49 μm (p < 0.001), 508 ± 38 μm (p < 0.001), 512 ± 38 μm (p < 0.001), and 518 ± 35 μm (p < 0.001) at 1, 3, 6, and 12 months postoperatively, respectively. CCT was significantly thinner in DMEK eyes than in control eyes (531 ± 32 μm) at 3 and 6 months postoperatively, whereas it was similar to that in control eyes 12 months postoperatively.

Although PCT after DMEK significantly decreased throughout the postoperative period, PCT in DMEK eyes never normalized as compared to healthy controls. Preoperative PCT was 821 ± 106 μm and changed to 765 ± 54 μm (p < 0.001), 723 ± 43 μm (p < 0.001), 710 ± 39 μm (p < 0.001), and 704 ± 41 μm (p = 0.002) at 1, 3, 6, and 12 months postoperatively, respectively. All PCTs were significantly larger in DMEK eyes compared to those in control eyes (669 ± 38 μm) (p < 0.001).

## Keratometric values

Table 3 shows the time course of KV. Eyes after DMEK had comparable anterior and total KV compared to healthy controls, and these values did not change significantly after DMEK. Anterior KV changed from 49.1 ± 2.0 D preoperatively to 49.0 ± 1.6 D at 1 month (p = 0.695),

**Table 1. Patient characteristics.**

| Factors | DMEK eyes | Control eyes | P–Value |
|---|---|---|---|
| Number of eyes | 30 | 31 | |
| Ethnicity: Japanese (%) | 30 (100) | 31 (100) | |
| Sex: Male (%) | 6 (20) | 13 (41.9) | 0.097 |
| Eye: Right (%) | 17 (56.7) | 18 (58.1) | 0.912 |
| Age (years) (mean ± SD) | 73.4 ± 7.4 | 73.0 ± 6.7 | 0.829 |
| Etiology (n) | FECD (9)/PBK (21) | Healthy eyes | |

DMEK, Descemet membrane endothelial keratoplasty; SD, standard deviation; FECD, Fuchs endothelial corneal dystrophy; PBK, pseudophakic bullous keratopathy.

**Table 2. Time course of clinical values after DMEK.**

| Factors | DMEK eyes | Control eyes | P–Value |
|---|---|---|---|
| **BCVA (logMAR) (mean ± SD)** | | | |
| Preoperative | 0.87 ± 0.52 | -0.03 ± 0.03 | **< 0.001** * |
| Post-op month 1 | 0.26 ± 0.23 | | **< 0.001** * |
| Post-op month 3 | 0.12 ± 0.15 | | **< 0.001** * |
| Post-op month 6 | 0.05 ± 0.11 | | **0.024** * |
| Post-op month 12 | 0.04 ± 0.11 | | **0.001** * |
| **CCT (μm) (mean ± SD)** | | | |
| Preoperative | 682 ± 100 | 531 ± 32 | **< 0.001** * |
| Post-op month 1 | 523 ± 49 | | 0.598 |
| Post-op month 3 | 508 ± 38 | | **0.014** * |
| Post-op month 6 | 512 ± 38 | | **0.042** * |
| Post-op month 12 | 518 ± 35 | | 0.937 |
| **Peripheral CT at 9.0 mm (μm) (mean ± SD)** | | | |
| Preoperative | 821 ± 106 | 669 ± 38 | **< 0.001** * |
| Post-op month 1 | 765 ± 54 | | **< 0.001** * |
| Post-op month 3 | 723 ± 43 | | **< 0.001** * |
| Post-op month 6 | 710 ± 39 | | **< 0.001** * |
| Post-op month 12 | 704 ± 41 | | **< 0.001** * |

DMEK, Descemet membrane endothelial keratoplasty; BCVA, best corrected visual acuity; CCT, central corneal thickness; CT, corneal thickness; Post-op, Postoperative.

**Table 3. Time course of keratometric values after DMEK.**

| Factors | DMEK eyes | Control eyes | P–Value |
|---|---|---|---|
| Anterior KV (D) (mean ± SD) | | | |
| **Preoperative** | 49.1 ± 2.0 | 49.4 ± 1.4 | 0.863 |
| **Post-op month 1** | 49.0 ± 1.6 | | 0.559 |
| **Post-op month 3** | 49.0 ± 1.7 | | 0.559 |
| **Post-op month 6** | 49.4 ± 1.2 | | 0.724 |
| **Post-op month 12** | 49.4 ± 1.6 | | 0.914 |
| Posterior KV (D) (mean ± SD) | | | |
| **Preoperative** | -6.6 ± 0.5 | -6.3 ± 0.2 | **0.018** * |
| **Post-op month 1** | -6.6 ± 0.4 | | **<0.001** * |
| **Post-op month 3** | -6.5 ± 0.3 | | **<0.001** * |
| **Post-op month 6** | -6.5 ± 0.3 | | **0.024** * |
| **Post-op month 12** | -6.4 ± 0.3 | | **0.023** * |
| Total KV (D) (mean ± SD) | | | |
| **Preoperative** | 42.8 ± 1.9 | 43.2 ± 1.3 | 0.516 |
| **Post-op month 1** | 42.5 ± 1.5 | | 0.105 |
| **Post-op month 3** | 42.9 ± 1.4 | | 0.105 |
| **Post-op month 6** | 43.1 ± 1.0 | | 0.395 |
| **Post-op month 12** | 43.1 ± 1.4 | | 0.584 |

DMEK, Descemet membrane endothelial keratoplasty; KV, keratometric value; Post-op, Postoperative.

* Mann–Whitney U test.

49.0 ± 1.7 D at 3 months (p = 0.695), 49.4 ± 1.2 D at 6 months (p = 0.918), and 49.4 ± 1.6 D at 12 months (p = 0.853) postoperatively. Total KV changed from 42.8 ± 1.9 D preoperatively to 42.5± 1.5 D at 1 month (p = 0.679), 42.9 ± 1.4 D at 3 months (p = 0.679), 43.1 ± 1.0 D at 6 months (p = 0.679), and 43.1 ± 1.4 D at 12 months (p = 0.600) postoperatively.

Eyes after DMEK had steeper posterior KV compared to healthy controls. DMEK surgery resulted in a slight decrease in posterior KV, but the value did not normalize compared to healthy controls. Posterior KV changed from -6.6 ± 0.5 D preoperatively to -6.6 ± 0.4 D at 1 month (p = 0.589), -6.5 ± 0.3 D at 3 months (p = 0.589), -6.5 ± 0.3 D at 6 months (p = 0.395), and -6.4 ± 0.3 D at 12 months (p = 0.351) postoperatively.

## HOAs

The time course of HOAs is shown in Fig 1. Anterior HOAs after DMEK improved from 1.92 ± 1.20 μm preoperatively to 1.62 ± 0.64 μm at 1 month (p = 0.723), 1.38 ± 0.71 μm at 3 months (p = 0.095), 1.22 ± 0.49 μm at 6 months (p = 0.033), and 1.03 ± 0.41 μm at 12 months (p < 0.001) postoperatively. Posterior HOAs after DMEK improved from 0.56 ± 0.28 μm preoperatively to 0.51 ± 0.23 μm at 1 month (p = 0.438), 0.44 ± 0.23 μm at 3 months (p = 0.069), 0.36 ± 0.17 μm at 6 months (p = 0.002), and 0.30 ± 0.13 μm at 12 months (p < 0.001) postoperatively. Total HOAs after DMEK improved from 1.94 ± 1.17 μm preoperatively to 1.62 ± 0.59 μm at 1 month (p = 0.734), 1.38 ± 0.63 μm at 3 months (p = 0.080), 1.23 ± 0.47 μm at 6 months (p = 0.0017), and 1.05 ± 0.42 μm at 12 months (p < 0.001) postoperatively. DMEK eyes showed greater values for all HOAs (p < 0.001) at all postoperative time points.

Anterior Comas after DMEK improved from 1.53 ± 1.04 μm preoperatively to 1.27 ± 0.54 μm at 1 month (p = 0.728), 1.12 ± 0.64 μm at 3 months (p = 0.204), 1.01 ± 0.44 μm at 6 months (p = 0.075), and 0.88 ± 0.39 μm at 12 months (p = 0.005) postoperatively. Posterior Comas after DMEK improved from 0.45 ± 0.26 μm preoperatively to 0.38 ± 0.20 μm at 1 month (p = 0.391), 0.32 ± 0.19 μm at 3 months (p = 0.066), 0.26 ± 0.15 μm at 6 months (p = 0.004), and 0.21 ± 0.13 μm at 12 months (p < 0.001)postoperatively. Total Comas after

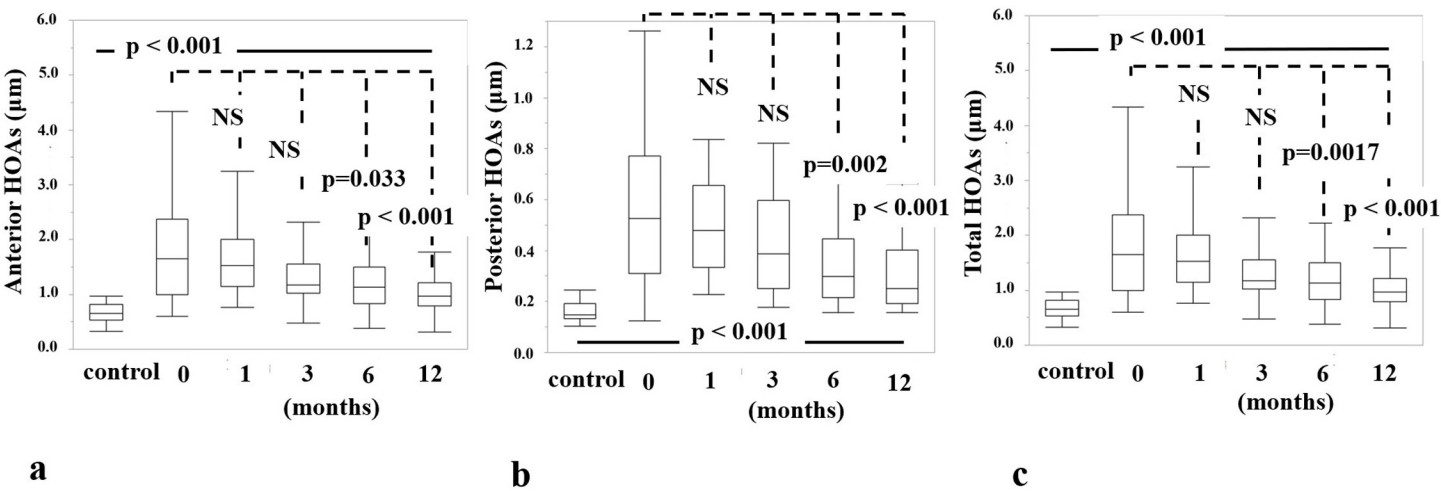

**Fig 1. Time course of Higher-Order Aberrations (HOAs).** (a) Anterior HOAs after DMEK decreased from 1.92 ± 1.20 μm preoperatively to 1.62 ± 0.64 μm at 1 month (p = 0.723), 1.38 ± 0.71 μm at 3 months (p = 0.095), 1.22 ± 0.49 μm at 6 months (p = 0.033), and 1.03 ± 0.41 μm (p < 0.001) at 12 months postoperatively. (b) Posterior HOAs after DMEK changed from 0.56 ± 0.28 μm preoperatively to 0.51 ± 0.23 μm at 1 month (p = 0.438), 0.44 ± 0.23 μm at 3 months (p = 0.069), 0.36 ± 0.17 μm at 6 months (p = 0.002), and 0.30 ± 0.13 μm (p < 0.001) at 12 months postoperatively. (c) Total HOAs after DMEK decreased from 1.94 ± 1.17 μm preoperatively to 1.62 ± 0.59 μm at 1 month (p = 0.734), 1.38 ± 0.63 μm at 3 months (p = 0.080), 1.23 ± 0.47 μm at 6 months (p = 0.0017), and 1.05 ± 0.42 μm at 12 months (p < 0.001) postoperatively. Although there was no significant improvement at 1 month or 3 months postoperatively, all factors significantly decreased after 6 months. Data are presented as the mean ± standard deviation. Abbreviations: NS, no significant difference; DMEK, Descemet membrane endothelial keratoplasty.

DMEK improved from 1.55 ± 0.97 μm preoperatively to 1.29 ± 0.53 μm at 1 month
(p = 0.706), 1.13 ± 0.59 μm at 3 months (p = 0.084), 1.03 ± 0.44 μm at 6 months (p = 0.033),
and 0.88 ± 0.39 μm at 12 months (p = 0.002) postoperatively (S1 Fig in S1 File). DMEK eyes
showed greater values for all Comas (p < 0.001) at all postoperative time points.

Anterior SAs after DMEK improved from 1.11 ± 0.70 μm preoperatively to 0.96 ± 0.45 μm
at 1 month (p = 0.825), 0.76 ± 0.40 μm at 3 months (p = 0.043), 0.66 ± 0.30 μm at 6 months
(p = 0.005), and 0.55 ± 0.23 μm at 12 months (p < 0.001) postoperatively. Posterior SAs after
DMEK changed from 0.32 ± 0.14 μm preoperatively to 0.33 ± 0.12 μm at 1 month (p = 0.231),
0.29 ± 0.13 μm at 3 months (p = 0.284), 0.24 ± 0.09 μm at 6 months (p = 0.017), and
0.21 ± 0.07 μm at 12 months (p < 0.001) postoperatively. Total SAs after DMEK improved
from 1.11 ± 0.70 μm preoperatively to 0.94 ± 0.39 μm at 1 month (p = 0.941), 0.75 ± 0.39 μm at
3 months (p = 0.057), 0.64 ± 0.28 μm at 6 months (p = 0.004), and 0.55 ± 0.25 μm at 12 months
(p < 0.001) postoperatively (S2 Fig in S1 File). Almost all SAs were higher in DMEK eyes than
in control eyes (p < 0.001) at all postoperative time points, except the anterior SAs at 12
months (p = 0.077) postoperatively.

### Correlations between visual acuity and aberration factors

There were no correlations between the final BCVA (12 months postoperatively) and the aber-
ration factors at any postoperative point (all, p > 0.05). Results are shown in S1 Table.

All available data are included in S2 Table (dataset).

## Discussion

This study prospectively examined visual outcomes and corneal characteristics in eyes under-
going DMEK throughout the first postoperative year. After DMEK, all eyes experienced an
improvement in BCVA and a reduction in HOAs; however, none of them reached a level com-
parable to healthy controls. Both CCT and PCT decreased after DMEK, with CCT normalizing
but PCT remaining thicker compared to that of controls. Eyes after DMEK had steeper poste-
rior KV compared to that of healthy controls; this value did not normalize even at 12 months
after surgery.

A thorough consideration of the corneal curvature is important to explain these findings.
During DMEK, the central endothelium is stripped and replaced with an under-sized graft
containing healthy endothelial cells. Post-operatively, these transplanted cells may migrate
into the periphery to fill the area between the edge of the graft and the area of stripping, and
come to a halt at the native peripheral endothelium due to contact inhibition.

Interestingly, Fig 2B shows the typical distribution of endothelial cells after DMEK, which
is different from that in healthy corneas (Fig 2A). In DMEK, stripping of the host endothelium
with Descemet's membrane (descemetorhexis) should be either the same or larger than the
graft size to prevent graft detachment. Since the peripheral endothelial function is reduced,
corneal edema may occur more frequently in the peripheral area (Fig 2C). In fact, our results
showed that PCT is significantly larger in DMEK patients. After DMEK, the center area with
the DMEK graft should show rapid improvement of corneal edema. However, as shown in Fig
2C, a residual edema in the peripheral area could be detected even after non-eventful DMEK.
This results in corneal irregularity in DMEK eyes compared to normal eyes, especially for the
theoretical SAs. As shown in Fig 3, the corneal curvature evidently differs in DMEK corneas
and healthy corneas. The posterior surface shifts forward, and the central cornea shows thin-
ning. Although it is uncertain whether the corneal edema could improve beyond the edge of
the graft, there is a hypothesis that transplanted endothelial cells could migrate to the periph-
eral area without the Descemet membrane [27]. Since all measurement points of the PCT (9.0

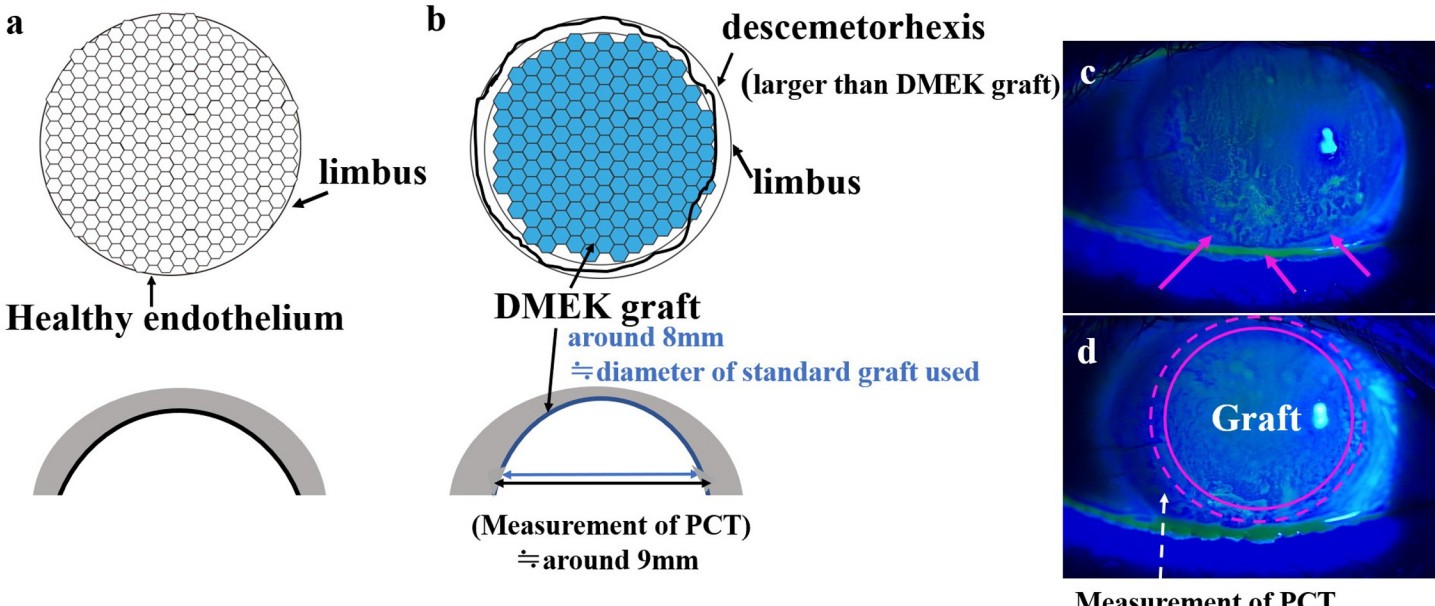

**Fig 2. Comparison between DMEK eyes and healthy eyes.** In comparison with a healthy cornea (a), DMEK eyes (b) contain few healthy endothelia in the peripheral cornea, because the graft size (around 8.0 mm) tends to be smaller than the stripping area (around 9.0 mm). (Blue line indicates the transplanted DMEK graft. Black line shows the measurement point of PCT [around 9.0 mm].) At the early phase after DMEK [within 1 month], peripheral edema was occasionally detected in the peripheral cornea as bullae (c) or as an irregular epithelium (d). (Arrows show corneal epithelial edema in the peripheral cornea, where DMEK graft does not cover. Circle indicates an image of transplanted graft. Broken circle and arrow indicate the measurement point of PCT [around 9.0 mm]). Abbreviations: DMEK, Descemet membrane endothelial keratoplasty. PCT, peripheral corneal thickness.

mm) were always along the outer rather than the edge of the graft, we could not evaluate the direct effect after DMEK. This problem could be one limitation of this study.

Since DMEK eyes may occasionally show central thinning [15, 28, 29], we also evaluated the corneal thickness in peripheral cornea at a 9.0-mm diameter (PCT). We found that both CCT and PCT gradually improved postoperatively, and CCT returned to normal thickness at 12 months, which indicated that corneal thickness remodeling occurred over time after DMEK. This might be attributable to migration of corneal endothelial cells to the peripheral area.

The center-to-periphery discrepancy in cell density may be especially highlighted in this patient population of largely pseudophakic bullous keratopathy. Whereas Fuchs endothelial dystrophy preferentially affects central endothelium, PBK is thought to result in a more global loss of endothelial cells. In fact, in our study, 21 of 30 DMEK cases (70%) were PBK. The high percentage of PBK in our cohort may contribute to the findings that PCT did not normalize after surgery. It is noteworthy that corneal HOAs were reduced at 6 to 12 months after DMEK, which coincided with improvement of PCT at 12 months. Thus, we consider that the assessment of PCT would be a clinically relevant parameter when evaluating improvement in corneal edema after DMEK.

Careful consideration is essential to understand specific corneal characteristics after DMEK. Although excellent visual outcome such as 20/20 vision should be expected after DMEK, the visual outcome is not completely equivalent to that in healthy controls. In fact, our study strongly indicates the presence of higher aberration factors in DMEK eyes, which suggests an inferiority to normal eyes.

This study allows for the following conclusions to be drawn: specific refractive changes after DMEK may be a result of the irregular distribution of the center-biased endothelium, and

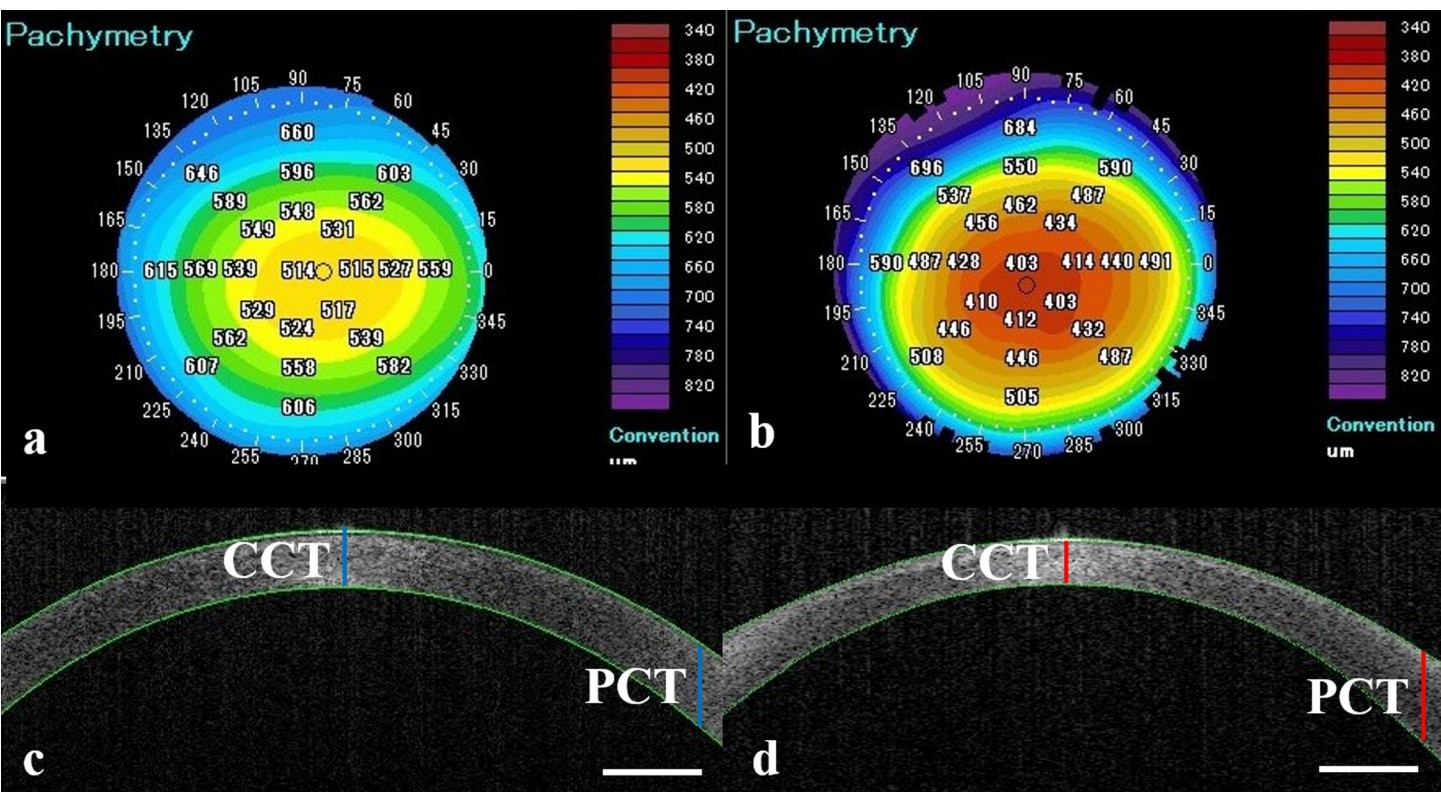

**Fig 3. Differences in topography between DMEK eyes and healthy eyes.** Normal corneas from healthy controls shows normal thickness in the topography map (a) and maintains the parallelism of the anterior and posterior cornea in the cross section of the same patient (c). However, DMEK eyes occasionally show corneal thinning in the topography map (b), and the cross section shows an irregular surface, especially in the posterior cornea (d). White bar indicates 1.0 mm. Abbreviations: DMEK, Descemet membrane endothelial keratoplasty; CCT, central corneal thickness; PCT, peripheral corneal thickness.

consequently, refractive changes are specific to the posterior surface. Additionally, although DMEK can significantly improve endothelial dysfunction and aberration factors, the optical quality is inferior to that in healthy eyes.

The strength of this study lies in its prospective design and meticulous post-operative follow-up without any missing data. The limitations include an ethnically limited patient population (all Japanese) with high predominance of pseudophakic bullous keratopathy with epithelial changes, which may indicate more severe disease than a patient with Fuchs dystrophy. In addition, although many of our findings are statistically significant, the clinical significance of such small differences are unknown. Lastly, backscatter and corneal haze were not evaluated because AS-OCT is not equipped to measure backscatter or densitometry.

This study provides valuable information regarding the long-term post-operative outcomes after DMEK. While the clinical results have been overall excellent, and certainly superiorly to full-thickness PKP and DSAEK, even after 1 year post-operatively these eyes are not comparable to healthy controls. We hypothesize these persistent changes may be a result of an irregular center-to-peripheral distribution of endothelial cells.

## Supporting information

**S1 File.**
(PDF)

**S1 Table. Correlations between best spectacle corrected visual acuity at 12 months and aberration factors.**
(PDF)

**S2 Table. Dataset.**
(XLSX)

## Acknowledgments

The authors thank Takahiko Hayashi, Hidenori Takahashi, Itaru Oyakawa, Hideaki Yoko-gawa, Akira Kobayashi, Naoko Kato, and Hidetoshi Kawashima for their recruitment and treatment of the patients. No other affiliation played a role in this study. There is no commercial affiliation such as employment, consultancy, patents, products in development, or marketed products, to declare. Any commercial affiliation does not alter our adherence to all PLOS ONE policies on sharing data and materials by including the following statement.

## Author Contributions

**Data curation:** Hidenori Takahashi.

**Investigation:** Akira Kobayashi, Itaru Oyakawa, Naoko Kato.

**Supervision:** Takefumi Yamaguchi.

**Validation:** Takefumi Yamaguchi.

**Writing – review & editing:** Takahiko Hayashi, Akira Kobayashi, Hidenori Takahashi.

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
