## [Decision Letter · Decision Letter 0]

8 May 2020

PONE-D-20-11108

Optical characteristics after Descemet membrane endothelial keratoplasty: 1-year results

PLOS ONE

Dear Dr. Hayashi,

Thank you for submitting your manuscript to PLOS ONE. After careful consideration, we would like to invite you to submit a revised version of the manuscript that addresses the points raised during the review process.

We would appreciate receiving your revised manuscript by Jun 22 2020 11:59PM. To enhance the reproducibility of your results, we recommend that if applicable you deposit your laboratory protocols in protocols.io, where a protocol can be assigned its own identifier (DOI) such that it can be cited independently in the future. For instructions see: http://journals.plos.org/plosone/s/submission-guidelines#loc-laboratory-protocols

We look forward to receiving your revised manuscript.

Kind regards,

Yu-Chi Liu, M.D

Academic Editor

PLOS ONE

Journal Requirements:

2. Please provide additional details regarding participant consent.

In the ethics statement in the Methods and online submission information, please ensure that you have specified (i) whether consent was informed and (ii) what type you obtained (for instance, written or verbal, and if verbal, how it was documented and witnessed).

If your study included minors, state whether you obtained consent from parents or guardians.

If the need for consent was waived by the ethics committee, please include this information.

'The funders had no role in study design, data collection and analysis, decision to publish, or preparation of the manuscript. This work was supported by Alexander von Humboldt Foundation (https://www.humboldt-foundation.de/web/home.html) and the Japan Eye Bank Association (http://www.j-eyebank.or.jp), but did not have any additional role in the study design, data collection and analysis, decision to publish, or preparation of the manuscript.'

Please remove any funding-related text from the manuscript and let us know how you would like to update your Funding Statement. Currently, your Funding Statement reads as follows: "No."

4. Your ethics statement must appear in the Methods section of your manuscript. If your ethics statement is written in any section besides the Methods, please move it to the Methods section and delete it from any other section. Please also ensure that your ethics statement is included in your manuscript, as the ethics section of your online submission will not be published alongside your manuscript.

5. Thank you for including your ethics statement:

'This prospective study was approved by the Institutional Review Board (approval no. YKH_29_03_05) and was performed in accordance with the ethical standards as laid down in the 1964 Declaration of Helsinki and its later amendments. The study procedures followed all institutional guidelines, and all patients provided informed consent.'

Reviewers' comments:

Reviewer's Responses to Questions

**Comments to the Author**

1. Is the manuscript technically sound, and do the data support the conclusions?

Reviewer #1: No

Reviewer #2: Yes

Reviewer #3: Partly

2. Has the statistical analysis been performed appropriately and rigorously? 

Reviewer #1: No

Reviewer #2: Yes

Reviewer #3: I Don't Know

3. Have the authors made all data underlying the findings in their manuscript fully available?

Reviewer #1: Yes

Reviewer #2: Yes

Reviewer #3: Yes

4. Is the manuscript presented in an intelligible fashion and written in standard English?

Reviewer #1: No

Reviewer #2: Yes

Reviewer #3: No

5. Review Comments to the Author

Reviewer #1: General Comments

Concise manuscript that discusses the effects of DMEK on corneal curvature and HOAs. However, the methodology requires some clarification before this manuscript can be considered for publication. The main issue is the criteria and justification for the control group. Moreover, extensive editing for grammar and English is required throughout the manuscript.

Specific Comments

Methods: Please state the inclusion criteria for DMEK cases e.g. FED or PBK etc. and inclusion for criteria for ‘age-matched’ controls.

Methods: Why was 31 controls chosen instead of 30? What was definition of ‘healthy’ and what were criteria e.g. pseudophakic or phakic, myopic or refraction status etc. as these affect the corneal curvature.

Methods: For sample size, please explain what is ‘cataract controls’

Methods: Please describe how the aberrations were derived from ASOCT with matching figure of the output and how the machine derives the values. This is important as each system varies on how the values are derived.

Methods: Were all DMEK patients psuedophakic? Were all the controls psuedophakic? If so, please include the HOA reading for the controls before and after cataract surgery.

Reviewer #2: The authors provide excellent information about corneal characteristics throughout the first post-operative year after DMEK. Most of the suggestions below are minor. However, the entire Discussion section needs to be re-worked to provide a more cohesive analysis and discussion of limitations.

Full comments have been uploaded as an attached document

Reviewer #3: REVIEW FOR PONE-D-20-11108

1. Summary of the research and overall impression

The main research question was to evaluate the corneal shape after DMEK over 12 months and compare it to a control group. Claims included that posterior keratometric values (KV) of DMEK eyes did not compare as favorable as anterior ones to the control, that higher order aberrations (HOAs) and peripheral corneal thickness stayed significantly higher for DMEK eyes at 12 months than the control. The conclusion of the study was that DMEK eyes showed higher HOAs and more corneal irregularity. The claims are considered significant for the specialty and previous important literature was referenced. The optical quality of the cornea after DMEK was evaluated in two previous studies (1. Optical quality of the cornea after Descemet membrane endothelial keratoplasty. van Dijk K, Droutsas K, Hou J, Sangsari S, Liarakos VS, Melles GR Am J Ophthalmol. 2014 Jul;158(1):71-79.; 2. Corneal higher-order aberrations after Descemet's membrane endothelial keratoplasty. Ophthalmology. 2012 Mar;119(3):528-35. Rudolph M, Laaser K, Bachmann BO, Cursiefen C, Epstein D, Kruse FE). The current study fits into the literature concerning changes in HOA. Backscatter was not a parameter of this study.

The strengths of the manuscript are the postoperative follow-up over 12 months, the evaluation of relevant parameters (best-corrected visual acuity (BCVA), central corneal thickness, KV and HOA) and the number of eyes evaluated.

In order for the manuscript to proceed the authors should define the eyes receiving DMEK more precisely. The following questions should be addressed: “Were Triple-DMEKs included or excluded?“, “Was DMEK performed on only phakic, only pseudo-phakic or a mixed group of eyes?“. Another essential point for this manuscript is data collection and data interpretation for the parameter peripheral corneal thickness (PPT). More technical details should be added and clarification given. Otherwise one might have the impression that the authors missed something crucial. Up to my knowledge the company CorneaGen, having supplied the study center with DMEK tissues, sells grafts with the following diameters: 7.5 mm, 7.75 mm and 8.0 mm. Unfortunately, the used graft diameter was not explicitly stated. The reader might assume that it is equal to the descemetorhexis diameter of 8.0 mm. The PCT was measured at 9.0 mm. The difference should clearly be mentioned and consolidation given on data interpretation in the discussion section. Pictorial data should be chosen to give the reader an impression of the aforementioned technical details (edge of graft, point of measurement of PCT, diameter of standard graft used).

I consider this study to have sufficient potential and encourage the authors to re-submit a revised version. My overall recommendation is „Accept Manuscript after Revision“.

2. Discussion of specific areas for improvement

Major issues

The following essential points for this study should be addressed to let the manuscript proceed:

1. Define eyes receiving DMEK (as this might affect BCVA and/ or posterior corneal surface): Phakic, pseudo-phakic or mixed;

2. State the graft diameters used and discuss this in relation to the point of measurement of PCT. You might keep your data as it is but might want to explain that corneal edema is improving beyond the edge of the graft to some extent;

3. Depending on the used diameters data interpretation of PCT needs to be discussed;

4. Slit-lamp examination results might be addressed to mention corneal haze/ scarring (line 317-319) or other pathologies which may be present limiting BCVA;

5. The authors should clarify the discussion to avoid confusion (e.g. page 23, line 255-257: might this be caused by the difference between the edge of the graft and the 9-mm point of measurement of the PCT? ; page 23, line 258-259: the authors should specify the inconsistent findings; page 23, line 262-264: why inhomogeneous as this is a graft from a healthy donor cornea?; page 25, line 298-300: PCT affecting central BCVA?; page 25, line 306-308: explain “irregular distribution of the center-biased endothelium”);

6. Sample size and patient characteristics might be part of the Material and Methods section (page 9, line 146-162);

7. The authors state that cataract controls were used but report in their dataset for all 31 controls a logMAR-value between -0.08 to 0.0 (page 9, line 152 and dataset table [group “healthy control”; parameter BSCVA]). Clarification for this crucial point is needed.

Minor issues

Additional things to address to improve the overall manuscript whilst not affecting the conclusions:

1. Fix the uneven distribution of the tables over the word documents;

2. Please use the same abbreviations throughout the manuscript including tables. Keep in mind where to put the abbreviations in tables and avoid not abbreviating at all including supplemental material;

3. Use pictorial data which better matches the statements or simply improved illustrations (e.g.: Fig. 2b: the “inhomogeneous distribution” of endothelial cells is not clearly seen, Fig. 2c: peripheral edema is blurred or out-of-focus, Fig. 3a-d: irregular surfaces – whether anterior or posterior – cannot be seen clearly in either cross section). The authors might want to use a higher magnification, depict millimeter units to scale and identify the point of PCT measurement in relation to the edge of the graft;

4. Please avoid misplacing figure captions into main manuscript text sections. This is easy to avoid and makes a good impression for review;

5. Tables: Data seems to be missing for the control eyes at month 1, month 6 and month 12 while giving p-values for the respective follow-ups. The authors might adjust the layout to avoid confusion. N may be added for number of females/ males and number of eyes.

3. Any other points

The authors made a great effort in data collection (number of parameters collected over 12 months of follow-up), although I am quite amazed that no data points are missing. Original data is made available and sufficient information is given to reproduce the study. Non-specialists are able to comprehend the idea behind this study. The conclusions do not overreach. Limitations of the study are discussed. Advance in the field is judged not applicable. I have no concerns about ethics or plagiarism but a professional statistician should evaluate the numbers/ data. I might be available for PLOSONE to look at a revised version of the manuscript.

6. PLOS authors have the option to publish the peer review history of their article (what does this mean?). If published, this will include your full peer review and any attached files.

Reviewer #1: No

Reviewer #2: No

Reviewer #3: No

---

## [Author Response · Author response to Decision Letter 0]

20 May 2020

Reviewer #1: General Comments

Concise manuscript that discusses the effects of DMEK on corneal curvature and HOAs. However, the methodology requires some clarification before this manuscript can be considered for publication. The main issue is the criteria and justification for the control group. Moreover, extensive editing for grammar and English is required throughout the manuscript.

Thank you very much for your review of our work. Your comments have aided us in significantly improving our manuscript. We have re-verified grammar and English of our manuscript which has been edited by a native English speaker. (https://www.editage.jp/).

Specific Comments

Methods: Please state the inclusion criteria for DMEK cases e.g. FED or PBK etc. and inclusion for criteria for ‘age-matched’ controls.

Our inclusion criteria were as follows: pseudophakic eyes without stromal scarring, but with stromal edema were included. Therefore, this study included 9 FED and 21 PBK, as shown in Table 1 (We did not find any differences in patients’ demographics, postop visual acuity and corneal HOAs between FED and PBK). Phakic DMEK or triple DMEK (combined with cataract surgery) were excluded. Fuchs keratopathy without corneal edema, but with corneal guttae not included in the study. Age-matched phakic eyes were selected as healthy controls.

Following your suggestion, we have added the description as follows:

Our inclusion criteria were pseudophakic eyes without stromal scarring, but with stromal edema. Phakic DMEK or triple DMEK (combined with cataract surgery) were excluded because of the difference in procedure. (Line 106–108)

Age-matched phakic eyes without history of ocular surgery or ocular surface disease were selected as healthy controls. (Line 111–112)

Methods: Why was 31 controls chosen instead of 30? What was definition of ‘healthy’ and what were criteria e.g. pseudophakic or phakic, myopic or refraction status etc. as these affect the corneal curvature.

Thank you for these questions. The number of DMEK patients from July 2017 to Mar 2018 was 31; therefore we selected 31 phakic controls. However, one of DMEK patients dropped out from our study protocol, and this is why we analyzed 30 DMEK and 31 healthy controls. We have added the following description of healthy eyes:

“Age-matched phakic eyes without history of ocular surgery or ocular surface disease were selected concurrently as healthy controls.” (Line 111–112) 

Methods: For sample size, please explain what is ‘cataract controls’

Thank you for pointing this out. Unfortunately, this is a mistake. We have revised it to "phakic controls" as follows:

Thirty-one phakic controls were obtained and compared with DMEK patients during the same period. (Line 157-158)

Methods: Please describe how the aberrations were derived from ASOCT with matching figure of the output and how the machine derives the values. This is important as each system varies on how the values are derived.

Thank you for this comment. As shown in our previous paper, aberrations can be easily calculated by ASOCT (CASIA):

Sixteen rotating AS-OCT scans were used to reconstruct three-dimensional models of the entire corneal structure. The Casia SS-1000 system corrected distortions in the AS-OCT images based on the refractive index of the anterior surface. A corneal specialist (TH) carefully checked all AS-OCT images to ensure that the surface digitalization recognized by the automated inbuilt software was correct. Zernike coefficients were calculated using Zernike analysis as previously reported. In brief, the anterior and posterior corneal surfaces were reconstructed as a three-dimensional model from the corneal height data. The anterior, posterior, and total corneal aberrations at diameters of 4.0 mm and 6.0 mm were calculated separately using the installed ray tracing software (version 5.1). The refractive indices of the cornea and aqueous humor were set to 1.376 and 1.336, respectively. The wavefront aberration was expanded with normalized Zernike polynomials up to the 8th order. HOA was defined as the root mean square (RMS) of the 3rd to 8th order Zernike coefficients as follows:

SA was defined as the RMS of Z40 (spherical aberration) and Z60 (secondary spherical aberration). Coma was defined as the RMS of Z3-1 and Z31.

Methods: Were all DMEK patients psuedophakic? Were all the controls psuedophakic? If so, please include the HOA reading for the controls before and after cataract surgery.

Thank you for this question. All DMEK patients were pseudophakic, and healthy controls were phakic without cataract. We have revised the manuscript as follows:

“Our inclusion criteria were pseudophakic eyes without stromal scarring, but with stromal edema. Phakic DMEK or triple DMEK (combined with cataract surgery) were excluded because of the difference in procedure.” (Line 106–108)

“Age-matched phakic eyes without history of ocular surgery or ocular surface disease were selected concurrently as healthy controls” (Line 111–112)

Reviewer #2: The authors provide excellent information about corneal characteristics throughout the first post-operative year after DMEK. Most of the suggestions below are minor. However, the entire Discussion section needs to be re-worked to provide a more cohesive analysis and discussion of limitations.

Full comments have been uploaded as an attached document

Thank you very much for your review of our work. Your encouraging comments have aided us in significantly improving our manuscript.

Abstract

- Line 45: would change “corneal shape” to “corneal characteristics”

Following your suggestion, we have revised the text as follows:

“To evaluate the corneal characteristics after Descemet membrane endothelial keratoplasty (DMEK) compared with normal corneas.” (Line 45–46)

- Methods need to make it clear at what time points patients were evaluated. “Patients who underwent DMEK at Yokohama Minami Kyosai Hospital were included and evaluated pre-operatively and at post-operative months 1, 3, 6, and 12 and compared to healthy controls. Corneal characteristics evaluated included (…)”

Following your suggestion, we have revised the manuscript as follows:

Patients who underwent DMEK at Yokohama Minami Kyosai Hospital were included and prospectively evaluated pre-operatively and at post-operative months 1, 3, 6, and 12, and compared to healthy controls. Corneal characteristics evaluated included corneal curvature (keratometric value [ KV]; D), central corneal thickness (CCT), peripheral corneal thickness (PCT), and corneal higher-order aberrations [HOAs] at 6.0 mm diameter, calculated by anterior segment optical coherence tomography and logarithm of the minimal angle of resolution [logMAR]. (Line 47–53)

- Line 52: “Thirty eyes” should be “30 eyes”

Following your suggestion, we have revised the description as follows: 

“A total of 30 eyes of 30 patients” (Line 54)

- Line 61: change to “Despite achieving”

Following your suggestion, we have revised the description as follows:

“Despite achieving good visual function and excellent corneal clarity, ” (Line 63)

- Line 62: change to “showed a steeper posterior KV and higher corneal HOA’s”

Following your suggestion, we have revised the description as follows:

“Despite achieving good visual function and excellent corneal clarity, eyes that underwent DMEK showed a steeper posterior KV and higher corneal HOAs than normal eyes even at 12 months after surgery.” (Line 63–65)

Introduction

- Line 67: Dr. Eduard Zirm

Following your suggestion, we have revised the description as follows:

“Dr. Eduard Zirm” (Line 69)

- Prevalence of Fuchs dystrophy needs context (in the U.S.? Japan? Worldwide?)

Thank you for pointing this out. We meant worldwide.

Following your suggestion, we have added the description as follows:

“In the worldwide population over the age of 40 years, the incidence of Fuchs endothelial corneal dystrophy (FECD), which requires corneal transplantation, is approximately 5%” (Line 71–73)

- Would re-write first two paragraphs to focus more on differences of DSAEK vs. DMEK, as opposed to differences of PKP vs. DMEK

Following your suggestion, we have revised the description as follows:

“

Therefore, endothelial keratoplasty (EK), which includes Descemet’s stripping automated endothelial keratoplasty (DSAEK] and Descemet’s membrane endothelial keratoplasty (DMEK), has been introduced worldwide and found to have advantages over PKP in terms of astigmatism, risk of graft rejection, and visual recovery [4-6].

Especially, the minimally invasive surgical procedure DMEK, developed by Dr. Melles in 2006 [7], involves the selective exchange of a complex of Descemet’s membrane and the endothelial layer. DMEK offers two primary advantages: 1) rapid visual recovery with a better final visual outcome than other keratoplasty techniques [8, 9], and 2) an extremely low incidence of graft rejection even when compared to DSAEK [10-12]. Previous studies have proved the superiority of DMEK to either DSAEK or ultra-thin DSAEK in terms of visual function [13-17].” (Line 77–87)

- Lines 86-88: Would delete this sentence as it is an overly broad statement that irregularities in topography correlate with vision. Would keep the focus on studies examining irregularities in post-DMEK eyes

Following your suggestion, we have deleted the following description:

“We have already reported that topographic changes and improvements in irregularity after keratoplasty or corneal disease show a strong correlation with visual function [13, 15-21]. 

As we have already shown, the visual acuity may occasionally not reach 20/20. Moreover, irregular topographic patterns may appear even after DMEK.”

- Lines 89-90: Delete or incorporate into previous paragraph about prior studies on DMEK outcomes

Following your suggestion, we have deleted the following description: 

“As we have already shown, the visual acuity may occasionally not reach 20/20. Moreover, irregular topographic patterns may appear even after DMEK. “

Materials and Methods

- Study design: how were phakic patients with Fuchs handled? Phakic DMEK, staged, triples?

Our inclusion criteria were as follows: pseudophakic eyes without stromal scarring, but with stromal edema. Phakic DMEK or triple DMEK (combined with cataract surgery) were excluded because of the difference in procedure.

- Study design: were patients with sub-epithelial scarring from chronic/severe Fuchs included?

Our inclusion criteria were as follows: pseudophakic eyes without stromal scarring, but with stromal edema. Phakic DMEK or triple DMEK (combined with cataract surgery) were excluded because of the difference in procedure.

- Later in the paper you state that all eyes undergoing DMEK had epithelial changes such as bullae or microcystic edema. Does this imply that eyes with visually significant guttae were not candidates for DMEK?

Our inclusion criteria were as follows: pseudophakic eyes without stromal scarring, but with stromal edema. Phakic DMEK or triple DMEK (combined with cataract surgery) were not included. Fuchs keratopathy without corneal edema, but with corneal guttae was not included in our study.

Following your suggestion, we have revised the text as below;

“Our inclusion criteria were pseudophakic eyes without stromal scarring, but with stromal edema. Phakic DMEK or triple DMEK (combined with cataract surgery) were excluded because of the difference in procedure.” (Line 106–108)

- Line 110: define “properly-sized graft”

Thank you for pointing this out.

We always use a donor punch with a diameter of 7.75 mm, 8.0 mm, or 8.25 mm, and try to make the area of descemetorhexis at least 9.0 mm. However, the size of the descemetorhexis graft can change depending on the corneal curvature of the donor or host.

Following your suggestion, we have revised the description as follows:

“Briefly, after the creation of a descemetorhexis with an approximate diameter of 9.0-mm under air, an appropriately-sized graft (7.75 mm, 8.0 mm, or 8.25 mm) was inserted using an intraocular lens inserter (WJ-60M®; Santen, Osaka, Japan)” (Line 116–118)

In general, Descemet membrane was then gently peeled from the stroma to obtain a Descemet membrane/corneal endothelial cell flap that was 8 mm in diameter.

- Line 112: what % size bubble was left?

For fixation of the DMEK graft, the anterior chamber volume was filled with 20% sulfur hexafluoride (SF6) until 80% of the AC volume was filled.

Following your suggestion, we have revised the description as follows:

“Briefly, after the creation of a descemetorhexis with an approximate diameter of 9.0-mm under air, an appropriately-sized graft (7.75 mm, 8.0 mm, or 8.25 mm) was inserted using an intraocular lens inserter (WJ-60M®; Santen, Osaka, Japan), and was unfolded and fixated with 20% sulfur hexafluoride (SF6) until 80% of the anterior chamber volume was filled.” (Line 116–119)

- Line 120: “were evaluated preoperatively and at 1, 3, 6, and 12 months post-operatively”

Following your suggestion, we have revised the description.

- Line 127: delete “pre-op and 1, 3, 6, 12 months post-op” (redundant with earlier part of paragraph)

Following your suggestion, we have deleted the description.

- Lines 127-129: delete “Since DMEK eyes may occasionally show central thinning”. Would address this in the Discussion section instead

Following your suggestion, we have moved the description to the Discussion.

- Line 129-130: “healthy eyes with no history of ocular disease or surgery with the exception of cataracts”

Following your suggestion, we have removed the description, and revised the text as follows:

“Age-matched phakic eyes without history of ocular surgery or ocular surface disease were selected concurrently as healthy controls” (Line 111–112)

- Line 137: you list both “gender” and “sex”

Following your suggestion, we have removed “gender”. (Line 142)

Results

Sample size: this paragraph is confusing. Power calculations should be done BEFORE patient recruitment based on expected differences in primary outcome. How can you do a power calculation after-the-fact based on post-op year #1 HOA data?

Thank you for this question. We used our preliminary data (not published), and calculated the statistic power using these values. We are convinced that there was a similar tendency regarding the HOAs of DMEK eyes at one year and healthy controls.

- Line 157: delete the word “current”

Following your suggestion, we have removed the word “current”. (Line 161)

- Line 161: “All eyes showed epithelial disorders, such as bullae and microcystic changes”

 – Does this imply that DMEK was not performed for visually significant guttae? In this case, your population of patients undergoing DMEK may have more severe disease with implications for post-op remodeling, etc.

Thank you for this question; we fully agree with your opinion.

Our inclusion criteria were as follows: pseudophakic eyes without stromal scarring, but with stromal edema. Phakic DMEK or triple DMEK (combined with cataract surgery) were not included. Fuchs keratopathy without corneal edema, but with corneal guttae was not included in our study.

Table 1

- Change “numbers of eyes” to “number of eyes”

Following your suggestion, we have revised the text to “number of eyes”.

- Sex and eye: can just list % male and % right

Following your suggestion, we have revised this for clarity.

- Add data on ethnicity

Following your suggestion, we have added data on ethnicity.

Results – VA and corneal thickness

- I would suggest rearranging the presentation of data as follows:

“BCVA after DMEK significantly improved throughout the post-operative period… However BCVA at all time points was still inferior to that of normal controls. 

CCT after DMEK significantly decreased throughout the post-operative period…. And by post-op month 12 was comparable to normal controls.

PCT after DMEK significantly decreased throughout the post-operative period… However, PCT never normalized as compared to healthy controls”

We fully agree with you. Your suggestions were very valuable and useful to us.

Following your suggestions, we have revised the text as follows:

“Visual acuity and corneal thickness

BCVA significantly improved after DMEK throughout the post-operative period as follows: from 0.87 ± 0.52 preoperatively to 0.26 ± 0.23 at 1 month (Table 2, p < 0.001), 0.12 ± 0.15 at 3 months (p < 0.001), 0.05 ± 0.11 at 6 months (p < 0.001), and 0.04 ± 0.11 (p < 0.001) at 12 months postoperatively. However, BCVA was still inferior to that of normal controls at all time points.

CCT after DMEK significantly decreased throughout the post-operative period up to 6 months after surgery; at 12 months after surgery it was comparable to normal controls. CCT significantly decreased from 682 ± 100 μm preoperatively to 523 ± 49 μm (p < 0.001), 508 ± 38 μm (p < 0.001), 512 ± 38 μm (p < 0.001) and 518 ± 35 μm (p < 0.001) at 1, 3, 6, and 12 months postoperatively, respectively. CCT was significantly thinner in DMEK eyes than in control eyes (531 ± 32 μm) at 3 and 6 months postoperatively, whereas it was similar to that in control eyes 12 months postoperatively.

Although PCT after DMEK significantly decreased throughout the post-operative period, PCT never normalized as compared to healthy controls. Preoperative PCT was 821 ± 106 μm and changed to 765 ± 54 μm (p < 0.001), 723 ± 43 μm (p < 0.001), 710 ± 39 μm (p < 0.001), and 704 ± 41 μm (p = 0.002) at 1, 3, 6, and 12 months postoperatively, respectively. All PCTs were significantly larger in DMEK eyes compared to those in control eyes (669 ± 38 μm) (p < 0.001).” (Line 171–188)

Table 2

- Need to re-arrange columns/rows as follows:

 Control eyes DMEK eyes P-value

BCVA Pre-op: 

 POM #1: 

 POM #3: 

 POM #6: 

 POM #12: 

We fully agree with you. Your suggestion was very valuable and useful to us.

Following your suggestion, we have revised the Table 2 to improve its layout.

Keratometric values

- I would suggest rearranging the presentation of data as follows:

“Eyes after DMEK had comparable anterior and total KV compared to healthy controls and these values did not change significantly after DMEK. 

Eyes after DMEK had steeper posterior KV compared to healthy controls. DMEK surgery resulted in a slight decrease in posterior KV but did not normalize value compared to healthy controls”

We fully agree with you. Your suggestions were very valuable and useful to us.

Following your suggestion, we have revised the text as follows: 

Keratometric values

Table 3 shows the time course of KV. Eyes after DMEK had comparable anterior and total KV compared to healthy controls and these values did not change significantly after DMEK. Anterior KV changed from 49.1 ± 2.0 D preoperatively to 49.0 ± 1.6 D at 1 month (p = 0.695), 49.0 ± 1.7 D at 3 months (p = 0.695), 49.4 ± 1.2 D at 6 months (p = 0.918), and 49.4 ± 1.6 D at 12 months (p = 0.853) postoperatively. Total KV changed from 42.8 ± 1.9 D preoperatively to 42.5 ± 1.5 D at 1 month (p = 0.679), 42.9 ± 1.4 D at 3 months (p = 0.679), 43.1 ± 1.0 D at 6 months (p = 0.679), and 43.1 ± 1.4 D at 12 months (p = 0.600) postoperatively.

Eyes after DMEK had steeper posterior KV compared to healthy controls. DMEK surgery resulted in a slight decrease in posterior KV, but the value did not normalize compared to healthy controls. Posterior KV changed from -6.6 ± 0.5 D preoperatively to -6.6 ± 0.4 D at 1 month (p = 0.589), -6.5 ± 0.3 D at 3 months (p = 0.589), -6.5 ± 0.3 D at 6 months (p = 0.395), and -6.4 ± 0.3 D at 12 months (p = 0.351) postoperatively. Although there were no significant differences in the anterior or total KVs, the posterior KV was larger in DMEK eyes than in control eyes (p < 0.05). (Line 201–215)

Lines 189-191: Unclear what you mean by “The anterior curvature became flat temporally, followed by a steep change. The posterior curvature showed a steep change temporally and flattened 1 month post-operatively”. How was this measured quantitatively? Did this occur in all post-DMEK eyes?

Thank you for pointing this out. We agree with you that the sentence is confusing. 

We have, therefore, removed the sentence.

Discussion

- This whole section needs work

Summary paragraph suggestion:

“This study prospectively examined visual outcomes and corneal characteristics in eyes　undergoing DMEK throughout the first post-operative year. After DMEK, all eyes experienced an improvement in BCVA and a reduction in HOAs, however none reached a level that was comparable to healthy controls. CCT and PCT both decreased after DMEK, with CCT normalizing but PCT remaining elevated compared to controls. Finally, eyes with FECD had steeper posterior KV compared to healthy controls and this value did not normalize after DMEK.”

Thank you for this kind suggestion. Following your suggestion, we have revised the first paragraph of the discussion as follows:

“This study prospectively examined visual outcomes and corneal characteristics in eyes undergoing DMEK throughout the first post-operative year. After DMEK, all eyes experienced an improvement in BCVA and a reduction in HOAs; however, none of them reached a level that was comparable to healthy controls. Both central and peripheral corneal thickness (CCT and PCT) decreased after DMEK, with CCT normalizing but PCT remaining thicker compared to controls. Eyes after DMEK had steeper posterior KV compared to healthy controls and this value did not normalize even at 12 months after surgery.” (Line 273-279)

Possible mechanisms explaining above findings:

- Replacement of central endothelial cells but low cell density peripherally

- Posterior stroma is more easily hydrated compared to anterior stroma, thus edema due to FECD is more likely to affect the posterior KV

We fully agree with your suggestions. We have revised the discussion according to these suggestions. (Line 272–297)

- Would delete the Inoue et al reference unless you want to expand on the mechanism further – do you believe that the anatomy of the corneal nerves affects the location of corneal edema? 

We fully agree with your suggestion. Following your suggestion, we have deleted the relevant text and citation from the discussion. (Line 272–297)

Limitations:

- Line 311: “relatively small number of Asian participants” – weren’t all the patients recruited from a Japanese hospital? You need to report ethnicity in Table 1

Thank you for pointing this out. We have revised the text to clarify that all patients were Japanese ethnicity.

- Lines 312-314: this is not a limitation. Discussion of statistical power goes in the Methods section

Thank you for pointing this out. We agree that this part is not necessary in the discussion section.

Following your suggestion, this text was removed. We have briefly mentioned that statistical power calculation and appropriate sample size is a strength of the study.

- Lines 320-326: disagree with the rationale that you could not detect a correlation between HOAs and BCVA due to small intra-individual differences. If you were able to detect a statistically significant change in HOAs and BCVA at the various post-operative visits, then you should be able to calculate a correlation coefficient

We fully agree with you. Following your suggestion, we have deleted the discussion regarding DMEK vs DSAEK.

- Line 326-327: would delete this sentence on DMEK vs DSAEK as it is not discussed anywhere else in the paper

We fully agree with you. Following your suggestion, we have deleted the discussion regarding DMEK vs DSAEK.

- Would add the limitation that only eyes with epithelial changes (bullae or microcystic edema) were candidates for DMEK. Thus, your population of patients undergoing DMEK may have more severe disease compared to those studied in other papers (i.e. patients with guttae only)

- Would add limitation that even though some of your findings were statistically significant, there is a question about whether this translates to clinical significance (ex. a difference in posterior KV of -6.4 vs. -6.9)

We fully agree with you. Following your suggestion, we have revised the discussion as follows:

“This study had some strengths and limitations. The strengths include that we calculated the statistical power and used the appropriate sample size. Furthermore, our prospective study had no missing data. The study also had the following limitations. First, only Asian eyes (Japanese) with epithelial changes suggestive of more severe disease, such as bullae or microcystic edema, were included. Second, the clinical significance of some of our findings remains unknown, despite their statistical significance. Third, backscatter and corneal haze were not evaluated because AS-OCT is not equipped for backscatter or densitometry.” (Line 333–339)

Figure 2:

- Line 285 states that DMEK eyes show corneal thinning, however the corresponding image (Figure 2b) shows higher pachy values than Figure 2a

- Line 286 states that DMEK shows irregular posterior surface – I don’t think the resolution of the attached AS-OCT image supports that claim

Thank you for pointing this out. Following your suggestion, we have revised Figure 2.

Reviewer #3: REVIEW FOR PONE-D-20-11108

1. Summary of the research and overall impression

The main research question was to evaluate the corneal shape after DMEK over 12 months and compare it to a control group. Claims included that posterior keratometric values (KV) of DMEK eyes did not compare as favorable as anterior ones to the control, that higher order aberrations (HOAs) and peripheral corneal thickness stayed significantly higher for DMEK eyes at 12 months than the control. The conclusion of the study was that DMEK eyes showed higher HOAs and more corneal irregularity. The claims are considered significant for the specialty and previous important literature was referenced. The optical quality of the cornea after DMEK was evaluated in two previous studies (1. Optical quality of the cornea after Descemet membrane endothelial keratoplasty. van Dijk K, Droutsas K, Hou J, Sangsari S, Liarakos VS, Melles GR Am J Ophthalmol. 2014 Jul;158(1):71-79.; 2. Corneal higher-order aberrations after Descemet's membrane endothelial keratoplasty. Ophthalmology. 2012 Mar;119(3):528-35. Rudolph M, Laaser K, Bachmann BO, Cursiefen C, Epstein D, Kruse FE). The current study fits into the literature concerning changes in HOA. Backscatter was not a parameter of this study.

The strengths of the manuscript are the postoperative follow-up over 12 months, the evaluation of relevant parameters (best-corrected visual acuity (BCVA), central corneal thickness, KV and HOA) and the number of eyes evaluated.

In order for the manuscript to proceed the authors should define the eyes receiving DMEK more precisely. The following questions should be addressed: “Were Triple-DMEKs included or excluded?“, “Was DMEK performed on only phakic, only pseudo-phakic or a mixed group of eyes?“. Another essential point for this manuscript is data collection and data interpretation for the parameter peripheral corneal thickness (PPT). More technical details should be added and clarification given. Otherwise one might have the impression that the authors missed something crucial. Up to my knowledge the company CorneaGen, having supplied the study center with DMEK tissues, sells grafts with the following diameters: 7.5 mm, 7.75 mm and 8.0 mm. Unfortunately, the used graft diameter was not explicitly stated. The reader might assume that it is equal to the descemetorhexis diameter of 8.0 mm. The PCT was measured at 9.0 mm. The difference should clearly be mentioned and consolidation given on data interpretation in the discussion section. Pictorial data should be chosen to give the reader an impression of the aforementioned technical details (edge of graft, point of measurement of PCT, diameter of standard graft used).

I consider this study to have sufficient potential and encourage the authors to re-submit a revised version. My overall recommendation is „Accept Manuscript after Revision“.

Thank you very much for your review of our work. Your comments have aided us in significantly improving our manuscript. We have clarified the inclusion/exclusion criteria following the reviewer comments. 

2. Discussion of specific areas for improvement

Major issues

The following essential points for this study should be addressed to let the manuscript proceed:

1. Define eyes receiving DMEK (as this might affect BCVA and/ or posterior corneal surface): Phakic, pseudo-phakic or mixed;

Our inclusion criteria were as follows: pseudophakic eyes without stromal scarring, but with stromal edema. Phakic DMEK or triple DMEK (combined with cataract surgery) were excluded. Fuchs keratopathy without corneal edema, but with corneal guttae was not included in our study. Age-matched phakic eyes were selected as healthy controls.

Following your suggestion, we have added the description as follows:

“Our inclusion criteria were pseudophakic eyes without stromal scarring, but with stromal edema. Phakic DMEK or triple DMEK (combined with cataract surgery) were excluded because of the difference in procedure.” (Line 106–108)

2. State the graft diameters used and discuss this in relation to the point of measurement of PCT. You might keep your data as it is but might want to explain that corneal edema is improving beyond the edge of the graft to some extent;

We fully agree with you. We always use a donor punch with a diameter of 7.75mm, 8.0mm, or 8.25mm, and try to make the area of descemetorhexis at least 9.0 mm. However, the size of the descemetorhexis graft can change depending on the corneal curvature of donor or host.

Following your suggestion, we have revised the text as below:

“Briefly, after the creation of a descemetorhexis with an approximate diameter of 9.0-mm under air, an appropriately-sized graft (7.75 mm, 8.0 mm, or 8.25 mm) was inserted using an intraocular lens inserter (WJ-60M®; Santen, Osaka, Japan).” (Line 116–118)

In general, Descemet membrane was then gently peeled from the stroma at approximately 9 mm to obtain a Descemet membrane/corneal endothelial cell flap that was 8 mm in diameter.

Further, we have revised the Figure 3 to make it easier for readers to understand the manuscript.

2. Depending on the used diameters data interpretation of PCT needs to be discussed;

We always use a donor punch with a diameter of 7.75mm, 8.0mm, or 8.25mm, and try to make the area of descemetorhexis at least 9.0 mm. However, the size of the descemetorhexis graft can change depending on the corneal curvature of donor or host.

We have added the following text to the manuscript: 

As DMEK replaces only the central area of the back-layer of the cornea, it is possible that this results in high ECD in the central area, and low ECD in the peripheral area. Moreover, since the posterior stroma is more easily hydrated compared to the anterior stroma, the improvement of corneal edema after DMEK is more likely to affect the posterior KV than the anterior KV. (Line 281–285)

Further, we have revised the Figure 3 to make it easier for readers to understand the manuscript.

3. Slit-lamp examination results might be addressed to mention corneal haze/ scarring (line 317-319) or other pathologies which may be present limiting BCVA;

Following your suggestion, we have added the following text to the manuscript:

“Our inclusion criteria were pseudophakic eyes without stromal scarring, but with stromal edema. Phakic DMEK or triple DMEK (combined with cataract surgery) were excluded because of the difference in procedure.” (Line 106–109)

4. The authors should clarify the discussion to avoid confusion (e.g. page 23, line 255-257: might this be caused by the difference between the edge of the graft and the 9-mm point of measurement of the PCT? ; page 23, line 258-259: the authors should specify the inconsistent findings; page 23, line 262-264: why inhomogeneous as this is a graft from a healthy donor cornea?; page 25, line 298-300: PCT affecting central BCVA?; page 25, line 306-308: explain “irregular distribution of the center-biased endothelium”);

Thank you for these suggestions. As you suggested, the discussion was confusing, and we have revised the discussion as follows:

“This study prospectively examined visual outcomes and corneal characteristics in eyes undergoing DMEK throughout the first post-operative year. After DMEK, all eyes experienced an improvement in BCVA and a reduction in HOAs; however, none of them reached a level that was comparable to healthy controls. Both central and peripheral corneal thickness (CCT and PCT) decreased after DMEK, with CCT normalizing but PCT remaining thicker compared to controls. Eyes after DMEK had steeper posterior KV compared to healthy controls and this value did not normalize even at 12 months after surgery.

A thorough consideration of the corneal curvature is important to explain these findings. As DMEK replaces only the central area of the back-layer of the cornea, it is possible that this results in high ECD in the central area, and low ECD in the peripheral area. Moreover, since the posterior stroma is more easily hydrated compared to the anterior stroma, the improvement of corneal edema after DMEK is more likely to affect the posterior KV than the anterior KV.” (Line 273–285)

6. Sample size and patient characteristics might be part of the Material and Methods section (page 9, line 146-162);

Thank you for pointing this out. We agree that text regarding how to calculate statistic power, or how to include patients should be placed in the Methods, but how many patients were obtained after calculation, or observation should be placed in the Results.

7. The authors state that cataract controls were used but report in their dataset for all 31 controls a logMAR-value between -0.08 to 0.0 (page 9, line 152 and dataset table [group “healthy control”; parameter BSCVA]). Clarification for this crucial point is needed.

Thank you for pointing out this mistake. We have revised it to "phakic controls" as follows:

“Thirty-one phakic controls were obtained and compared with DMEK patients during the same period.” (Line 157–158)

Minor issues

Additional things to address to improve the overall manuscript whilst not affecting the conclusions:

1. Fix the uneven distribution of the tables over the word documents;

Thank you for pointing this out. Following your suggestion, we have revised the Tables.

2. Please use the same abbreviations throughout the manuscript including tables. Keep in mind where to put the abbreviations in tables and avoid not abbreviating at all including supplemental material;

We fully agree with your suggestion. Following your suggestion, we have revised the abbreviations in the manuscript.

3. Use pictorial data which better matches the statements or simply improved illustrations (e.g.: Fig. 2b: the “inhomogeneous distribution” of endothelial cells is not clearly seen, Fig. 2c: peripheral edema is blurred or out-of-focus, Fig. 3a-d: irregular surfaces – whether anterior or posterior – cannot be seen clearly in either cross section). The authors might want to use a higher magnification, depict millimeter units to scale and identify the point of PCT measurement in relation to the edge of the graft;

We fully agree with you. Following your suggestion, we have revised Figures 2 and 3.

3. Please avoid misplacing figure captions into main manuscript text sections. This is easy to avoid and makes a good impression for review;

We fully agree with you. Following your suggestion, we have revised the Figure captions so that they are inserted after the first mention of the figure in the text, as per the journal guidelines.

4. Tables: Data seems to be missing for the control eyes at month 1, month 6 and month 12 while giving p-values for the respective follow-ups. The authors might adjust the layout to avoid confusion. N may be added for number of females/ males and number of eyes.

Since controls are control eyes (no surgery), there is no postoperative data. Following the reviewers’ suggestion, we have revised the layout of the table.

Your suggestions were very valuable and useful to us.

Following your suggestion, we have revised Table 2.

3. Any other points

The authors made a great effort in data collection (number of parameters collected over 12 months of follow-up), although I am quite amazed that no data points are missing. Original data is made available and sufficient information is given to reproduce the study. Non-specialists are able to comprehend the idea behind this study. The conclusions do not overreach. Limitations of the study are discussed. Advance in the field is judged not applicable. I have no concerns about ethics or plagiarism but a professional statistician should evaluate the numbers/ data. I might be available for PLOSONE to look at a revised version of the manuscript.

Thank you for these comments. Regarding statistics, prof. Hidenori Takahashi has a qualification in statistics.

---

## [Decision Letter · Decision Letter 1]

20 Jun 2020

Dear Dr. Hayashi,

This is an automatic email. 

The PDF for your submission, "Optical characteristics after Descemet membrane endothelial keratoplasty: 1-year results" has been built. If you have not already done so, please review your manuscript and approve your PDF to complete your submission at https://www.editorialmanager.com/pone/. 

Thank you for your time and support.

PLOS ONE Staff

https://www.editorialmanager.com/pone/

---

## [Author Response · Author response to Decision Letter 1]

4 Jul 2020

Reviewer #2: Thank you for all your thoughtful responses and modifications. The revised manuscript is much more comprehensive and coherent. Most of my comments below are minor; however, the Discussion section still needs to be significantly re-worked.

Thank you very much for your review of our work. Your comments have helped us to significantly improve our manuscript.

Introduction

- Line 72: would modify, as not all patients with Fuchs requires corneal transplantation

- Would shorten paragraph about transition from PKP to EK

Following your suggestion, we have revised the description about Fuchs endothelial dystrophy and shortened paragraph about transition from PKP to EK. (Line 70-79)

- Dr. Gerrit Melles

Following your suggestion, we have revised the description as follow.

‘Dr. Gerrit Melles’ (Line 80)

Methods

- Line 117: modify to "appropriately under-sized graft"

Following your suggestion, we have revised the description as follow.

‘appropriately under-sized graft’ (Line 115)

Results

- Line 153-154: need to make clear that the data and calculations were based on non-published preliminary data

Following your suggestion, we have revised the description as follow.

‘From our calculations using unpublished preliminary data (the HOAs of DMEK eyes at one year and of control eyes were 0.97 ± 0.43 μm and 0.61 ± 0.19, respectively)’ (Line 151-152)

- Patient characteristics: Would move "All patients were of Japanese ethnicity" to line 164, after discussion of differences in age/sex

Following your suggestion, we have moved the patient description after description of age and sex. (Line 162)

- Line 213-215: Would delete, redundant with prior paragraphs

Following your suggestion, we have deleted the redundant description. 

- HOA's: After each paragraph, I would add a sentence comparing the DMEK outcomes to control eyes. Currently this comparison occurs in lines 260-264, but I think it would be easier for the reader if the comparison was done at the time each factor is discussed

Following your suggestion, we have replaced these descriptions. (Line 217-218, 243-244, and 253-255)

Discussion

- This whole section needs to be reworked so that it's more organized and coherent

- A few specific points below:

- Lines 302-303: was peripheral edema detected even at POM #12? If so, in how many eyes did this occur?

There was no eye with epithelial disorder such as bullae, or cystic change at 12 months after DMEK. We have revised the description as follows. 

‘At the early phase after DMEK [within 1 month], peripheral edema was occasionally detected in the peripheral cornea as bullae (c) or as an irregular epithelium (d).’ (Line 298-300)

- Lines 319-320: "which coincided with normalization of PCT at 12 months" - but I thought that PCT never normalized compared to controls?

Following your suggestion, we have replaced the description.

‘It is noteworthy that corneal HOAs were reduced at 6 to 12 months after DMEK, which coincided with improvement of PCT at 12 months.’ (Line 319-321)

- Lines 320-322: I don't think you can make this conclusion, especially since your analysis did not show correlation between any factor and VA

Following your suggestion, we have revised the description as follows.

‘Thus, we consider that the assessment of PCT would be a clinically relevant parameter when evaluating improvement in corneal edema after DMEK.’ (Line 321-322)

- Another point that should be added is the relatively high % of patients with PBK versus Fuchs. Patients with PBK likely have decreased reserve of healthy peripheral endothelium compared to Fuchs, which may contribute to your finding that PCT does not normalize

Following your suggestion, we have added a theorical comment in the discussion as follows.

‘In our study, 21 of 30 DMEK cases (70%) were PBK. The high percentage of PBK in our cohort may contribute to the findings that PCT did not normalize after surgery.’ (Line 318-319)

- Lines 333-334: Would delete sentence about statistical power

Following your suggestion, we have deleted the discussion about statistic power.

Table 2

- Would reformat columns so that it matches that of Table 3

Following your suggestion, we have reformatted Table 2.

Reviewer #3: REVIEW FOR PONE-D-20-11108

1. Overall impression

I still consider this study to have sufficient potential for publication. My overall recommendation is „Accept Manuscript after Revision“.

2. Open points

Cf. Revision 1:

State the graft diameters used and discuss this in relation to the point of measurement of PCT. You might keep your data as it is but might want to explain that corneal edema is improving beyond the edge of the graft to some extent;

Pictorial data should be chosen to give the reader an impression of the aforementioned technical details (edge of graft, point of measurement of PCT, diameter of standard graft used).

Unanswered or open points of major concern affecting the conclusions of the study/ concerning the methods used:

Unfortunately, the authors did not yet explicitly state that all measurements of the peripheral corneal thickness were at least 0.75-1.25 mm beyond the edge of the graft/ in diseased host cornea. Thus, the peripheral edema measured cannot be directly affected by DMEK surgery. This inevitably has effects on the interpretation of the data. Figure 2 does not show an inhomogenous distribution as stated in the manuscript text. The text should clearly refer to the addressed figure/ sub-figure.

The pictorial data needs to be adjusted accordingly. The cell density of the graft in figure 2 seems to be higher than of the healthy cornea. All points should be fixed in the schematic images. In addition, the used arrows and circle need to be explained to the reader.

Following your suggestion, we have revised the caption of Fig.2 and added the discussion as follows.

‘inhomogenous’ → ’typical’ (Line 277)

‘In comparison with a healthy cornea (a), DMEK eyes (b) contain few healthy endothelia in the peripheral cornea, because the graft size (around 8.0 mm) tends to be smaller than the stripping area (around 9.0 mm). (Blue line indicates the transplanted DMEK graft. Black line shows the measurement point of PCT [around 9.0 mm].) At the early phase after DMEK [within 1 month], peripheral edema was occasionally detected in the peripheral cornea as bullae (c) or as an irregular epithelium (d). (Arrows show corneal epithelial edema in the peripheral cornea, where DMEK graft does not cover. Circle indicates an image of transplanted graft. Broken circle and arrow indicate the measurement point of PCT [around 9.0 mm]).’ (Line 295-302)

‘Although it is uncertain whether the corneal edema could improve beyond the edge of the graft, there is a hypothesis that transplanted endothelial cells could migrate to the peripheral area without the Descemet membrane.27 Since all measurement points of the PCT (9.0 mm) were always along the outer rather than the edge of the graft, we could not evaluate the direct effect after DMEK. This problem could be one limitation of this study.’ (Line 288-292)

---

## [Decision Letter · Decision Letter 2]

27 Jul 2020

PONE-D-20-11108R2

Optical characteristics after Descemet membrane endothelial keratoplasty: 1-year results

PLOS ONE

Dear Dr. Hayashi,

Thank you for submitting your manuscript to PLOS ONE. We invite you to submit a revised version of the manuscript that addresses the points raised by the reviewer.

We look forward to receiving your revised manuscript.

Kind regards,

Yu-Chi Liu, M.D

Academic Editor

PLOS ONE

Reviewers' comments:

Reviewer's Responses to Questions

**Comments to the Author**

1. If the authors have adequately addressed your comments raised in a previous round of review and you feel that this manuscript is now acceptable for publication, you may indicate that here to bypass the “Comments to the Author” section, enter your conflict of interest statement in the “Confidential to Editor” section, and submit your "Accept" recommendation.

Reviewer #2: (No Response)

Reviewer #3: All comments have been addressed

2. Is the manuscript technically sound, and do the data support the conclusions?

Reviewer #2: Yes

Reviewer #3: Yes

3. Has the statistical analysis been performed appropriately and rigorously? 

Reviewer #2: Yes

Reviewer #3: I Don't Know

4. Have the authors made all data underlying the findings in their manuscript fully available?

Reviewer #2: Yes

Reviewer #3: Yes

5. Is the manuscript presented in an intelligible fashion and written in standard English?

Reviewer #2: Yes

Reviewer #3: Yes

6. Review Comments to the Author

Reviewer #2: Thank you for all your thoughtful responses and changes. Please see my suggestions below.

Abstract

-Line 57: delete comma after "mean age"

Introduction

- Would recommend shortening introductory paragraphs. May consider something as below:

"Corneal transplantation has evolved significantly since the first full-thickness keratoplasty was performed by Dr. Eduard Zirm in 1905. For patients with endothelial dysfunction, a partial-thickness endothelial keratoplasty has now become standard of care. In particular, Descemet's stripping endothelial keratoplasty (DMEK), developed by Dr. Gerrit Melles, has resulted in the best visual outcomes for these patients. In this procedure, the diseased endothelium and Descemet's membrane are replaced in an anatomically precise fashion."

Results

- Line 229: need space between "(p<0.001)postoperatively"

- Line 247: same as above

- Line 250: same as above

Discussion

- Line 269: need space between "controls;this"

- Lines 271 and beyond: I think this section still needs work to provide a clear explanation and narrative. The following paragraphs are suggestions only - please do not feel obligated to use. However I hope it demonstrates a clearer "flow" to this section.

"During DMEK, the central endothelium is stripped and replaced with an under-sized graft containing healthy endothelial cells. Post-operatively, these transplanted cells may migrate into the periphery to fill the area between the edge of the graft and the area of stripping, and come to a halt at the native peripheral endothelium due to contact inhibition.

This resultant discrepancy between high cell density in the center and lower cell density in the periphery may account for the persistent changes between the post-DMEK eye and a healthy control. As the central cornea deturgesces, the CCT normalizes but the PCT does not. Since the deturgescence occurs mostly via the posterior stroma, the posterior KV improves while the anterior KV remains unchanged.

This center-to-periphery discrepancy in cell density may be especially highlighted in this patient population of largely pseudophakic bullous keratopathy. Whereas Fuchs endothelial dystrophy preferentially affects central endothelium, PBK is thought to result in a more global loss of endothelial cells.

The strength of this study lies in its prospective design and meticulous post-operative follow-up without any missing data. The limitations include an ethnically limited patient population (all Japanese) with high predominance of pseudophakic bullous keratopathy with epithelial changes, which may indicate more severe disease than a patient with Fuchs dystrophy. In addition, although many of our findings are statistically significant, the clinical significance of such small differences are unknown. Lastly, backscatter and corneal haze were not evaluated because AS-OCT is not equipped to measure backscatter or densitometry.

This study provides valuable information regarding the long-term post-operative outcomes after DMEK. While the clinical results have been overall excellent, and certainly superiorly to full-thickness PKP and DSAEK, even after 1 year post-operatively these eyes are not comparable to healthy controls. We hypothesize these persistent changes may be a result of an irregular center-to-peripheral distribution of endothelial cells."

Reviewer #3: Dear authors,

Thank you for your responses and scientifically appropriate modifications. I hope that my review did significantly improve your manuscript. My overall recommendation to the Editor is "Accept".

7. PLOS authors have the option to publish the peer review history of their article (what does this mean?). If published, this will include your full peer review and any attached files.

Reviewer #2: No

Reviewer #3: No

---

## [Author Response · Author response to Decision Letter 2]

28 Jul 2020

Reviewer #2: Thank you for all your thoughtful responses and changes. Please see my suggestions below.

Thank you very much for your review of our work. Your comments have helped us to significantly improve our manuscript.

Abstract

-Line 57: delete comma after "mean age"

Following your suggestion, we have deleted the comma. (Line )

Introduction

- Would recommend shortening introductory paragraphs. May consider something as below:

"Corneal transplantation has evolved significantly since the first full-thickness keratoplasty was performed by Dr. Eduard Zirm in 1905. For patients with endothelial dysfunction, a partial-thickness endothelial keratoplasty has now become standard of care. In particular, Descemet's stripping endothelial keratoplasty (DMEK), developed by Dr. Gerrit Melles, has resulted in the best visual outcomes for these patients. In this procedure, the diseased endothelium and Descemet's membrane are replaced in an anatomically precise fashion."

Following your suggestion, we have revised the introduction as follow.

‘ Corneal transplantation has evolved significantly since the first full-thickness keratoplasty was performed by Dr. Eduard Zirm in 1905 [1]. However, because of injury, this technique is associated with disadvantages such as graft rejection, glaucoma (steroid-dependent) and suture-related problems, slow and low visual recovery with heavy astigmatism, infection, or global rupture [2]. For patients with endothelial dysfunction, a partial-thickness endothelial keratoplasty has now become standard of care [3-5]. 

In particular, Descemet's membrane endothelial keratoplasty (DMEK), developed by Dr. Gerrit Melles, has resulted in the best visual outcomes for these patients [6]. In this procedure, the diseased endothelium and Descemet's membrane are replaced in an anatomically precise fashion.’ (Line 70-79)

Results

- Line 229: need space between "(p<0.001)postoperatively"

Following your suggestion, we have revised the description as follow.

‘(p<0.001) postoperatively’ (Line 225-226)

- Line 247: same as above

Following your suggestion, we have revised the description as follow.

‘(p < 0.001) postoperatively’ (Line 244)

- Line 250: same as above

Following your suggestion, we have revised the description as follow.

‘(p < 0.001) postoperatively’ (Line 247)

Discussion

- Line 269: need space between "controls;this"

Following your suggestion, we have revised the description as follow.

‘controls; this’ (Line 266)

- Lines 271 and beyond: I think this section still needs work to provide a clear explanation and narrative. The following paragraphs are suggestions only - please do not feel obligated to use. However I hope it demonstrates a clearer "flow" to this section.

"During DMEK, the central endothelium is stripped and replaced with an under-sized graft containing healthy endothelial cells. Post-operatively, these transplanted cells may migrate into the periphery to fill the area between the edge of the graft and the area of stripping, and come to a halt at the native peripheral endothelium due to contact inhibition.

This resultant discrepancy between high cell density in the center and lower cell density in the periphery may account for the persistent changes between the post-DMEK eye and a healthy control. As the central cornea deturgesces, the CCT normalizes but the PCT does not. Since the deturgescence occurs mostly via the posterior stroma, the posterior KV improves while the anterior KV remains unchanged.

This center-to-periphery discrepancy in cell density may be especially highlighted in this patient population of largely pseudophakic bullous keratopathy. Whereas Fuchs endothelial dystrophy preferentially affects central endothelium, PBK is thought to result in a more global loss of endothelial cells.

The strength of this study lies in its prospective design and meticulous post-operative follow-up without any missing data. The limitations include an ethnically limited patient population (all Japanese) with high predominance of pseudophakic bullous keratopathy with epithelial changes, which may indicate more severe disease than a patient with Fuchs dystrophy. In addition, although many of our findings are statistically significant, the clinical significance of such small differences are unknown. Lastly, backscatter and corneal haze were not evaluated because AS-OCT is not equipped to measure backscatter or densitometry.

This study provides valuable information regarding the long-term post-operative outcomes after DMEK. While the clinical results have been overall excellent, and certainly superiorly to full-thickness PKP and DSAEK, even after 1 year post-operatively these eyes are not comparable to healthy controls. We hypothesize these persistent changes may be a result of an irregular center-to-peripheral distribution of endothelial cells."

Following your suggestion, we have revised the discussion. 

‘A thorough consideration of the corneal curvature is important to explain these findings. During DMEK, the central endothelium is stripped and replaced with an under-sized graft containing healthy endothelial cells. Post-operatively, these transplanted cells may migrate into the periphery to fill the area between the edge of the graft and the area of stripping, and come to a halt at the native peripheral endothelium due to contact inhibition. This resultant discrepancy between high cell density in the center and lower cell density in the periphery may account for the persistent changes between the post-DMEK eye and a healthy control. As the central cornea deturgesces, the CCT normalizes but the PCT does not. Since the deturgescence occurs mostly via the posterior stroma, the posterior KV improves while the anterior KV remains unchanged.’ (Line 268-276)

‘Since DMEK eyes may occasionally show central thinning [15, 27, 28], we also evaluated the corneal thickness in peripheral cornea at a 9.0-mm diameter (PCT). We found that both CCT and PCT gradually improved postoperatively, and CCT returned to normal thickness at 12 months, which indicated that corneal thickness remodeling occurred over time after DMEK. This might be attributable to migration of corneal endothelial cells to the peripheral area. The center-to-periphery discrepancy in cell density may be especially highlighted in this patient population of largely pseudophakic bullous keratopathy. Whereas Fuchs endothelial dystrophy preferentially affects central endothelium, PBK is thought to result in a more global loss of endothelial cells. In fact, in our study, 21 of 30 DMEK cases (70%) were PBK. The high percentage of PBK in our cohort may contribute to the findings that PCT did not normalize after surgery. It is noteworthy that corneal HOAs were reduced at 6 to 12 months after DMEK, which coincided with improvement of PCT at 12 months. Thus, we consider that the assessment of PCT would be a clinically relevant parameter when evaluating improvement in corneal edema after DMEK.’ (Line 313-326)

‘The strength of this study lies in its prospective design and meticulous post-operative follow-up without any missing data. The limitations include an ethnically limited patient population (all Japanese) with high predominance of pseudophakic bullous keratopathy with epithelial changes, which may indicate more severe disease than a patient with Fuchs dystrophy. In addition, although many of our findings are statistically significant, the clinical significance of such small differences are unknown. Lastly, backscatter and corneal haze were not evaluated because AS-OCT is not equipped to measure backscatter or densitometry.

This study provides valuable information regarding the long-term post-operative outcomes after DMEK. While the clinical results have been overall excellent, and certainly superiorly to full-thickness PKP and DSAEK, even after 1 year post-operatively these eyes are not comparable to healthy controls. We hypothesize these persistent changes may be a result of an irregular center-to-peripheral distribution of endothelial cells.’ (Line 337-348)

---

## [Decision Letter · Decision Letter 3]

18 Aug 2020

PONE-D-20-11108R3

Optical characteristics after Descemet membrane endothelial keratoplasty: 1-year results

PLOS ONE

Dear Dr. Hayashi,

Thank you for submitting your manuscript to PLOS ONE. The manuscript has been much improved. However, one of the reviewers has raised some comments. Therefore, we invite you to submit a revised version of the manuscript that addresses the points raised during the review process.

We look forward to receiving your revised manuscript.

Kind regards,

Yu-Chi Liu, M.D

Academic Editor

PLOS ONE

Reviewers' comments:

Reviewer's Responses to Questions

**Comments to the Author**

1. If the authors have adequately addressed your comments raised in a previous round of review and you feel that this manuscript is now acceptable for publication, you may indicate that here to bypass the “Comments to the Author” section, enter your conflict of interest statement in the “Confidential to Editor” section, and submit your "Accept" recommendation.

Reviewer #2: (No Response)

2. Is the manuscript technically sound, and do the data support the conclusions?

Reviewer #2: Yes

3. Has the statistical analysis been performed appropriately and rigorously? 

Reviewer #2: Yes

4. Have the authors made all data underlying the findings in their manuscript fully available?

Reviewer #2: Yes

5. Is the manuscript presented in an intelligible fashion and written in standard English?

Reviewer #2: Yes

6. Review Comments to the Author

Reviewer #2: Introduction

- Line 71: not sure what you mean by "because of injury". I would delete this. The sentence works without it: "However, this technique is associated with..."

- Line 73: replace "heavy" with "significant"

- Line 74: replace "global rupture" with "risk of globe rupture with trauma"

- Line 81: need to define DSAEK acronym

- Line 85: need space between "withcorneal"

Materials and Methods

- Line 106: would change "ocular surface disease" to "ocular disease". I presume patients with glaucoma or retinal pathology were excluded due to effect on VA

Results

- Line 160: need to define FECD acronym

Table 1

- Would use PBK acronym/definition instead of BK to keep consistent with text

Table 2/3

- Would make text consistent between "Post-operative month 1" and "1 month after surgery"

- Could also consider shortening to Post-op month 1 or POM #1 for ease of reading the table, and then defining the acronym in the table legend/description

Discussion

- Lines 268-327: these paragraphs have redundant/repeating elements - they all discuss discrepancy between central and peripheral endothelial cell density and how this may explain your results. The content of these paragraphs should be re-written so that each concept is discussed once and thoroughly.

"During DMEK, the central endothelium is stripped and replaced with an under-sized graft containing healthy endothelial cells. Post-operatively, these transplanted cells may migrate into the periphery to fill the area between the edge of the graft and the area of stripping, and come to a halt at the native peripheral endothelium due to contact inhibition. <<discuss 2="" diagram="" dmek="" figure="" graft="" here="" of="">>. This resultant discrepancy between high cell density in the center and low cell density in the periphery may account for the persistent changes between the post-DMEK eyes and healthy controls. As the central cornea deturgesces, the CCT normalizes but the PCT does not. <<discuss 2="" bullae="" figure="" here="" of="" peripheral="" photos="">>. Since the deturgescence occurs mostly via the posterior stroma, the posterior KV improves while the anterior KV remains unchanged. <<discuss 3="" and="" anterior="" between="" difference="" figure="" here="" in="" parallelism="" posterior="" showing="" surface="">>. <<discuss difference="" hoa="" how="" in="" results="" this="">> <<discuss explain="" fecd="" findings="" help="" high="" how="" in="" may="" of="" paper="" pbk="" proportion="" these="" versus="" your="">>".

Figure 2:

- Would modify (b) to more accurately reflect size of descemetorhexis and what you discuss in your text. A 9 mm descemetorhexis on an 11 mm cornea still leaves a peripheral population of native endothelial cells.

- Would maybe add an additional diagram showing how you hypothesize these centrally transplanted enothelial cells migrate to the periphery and result in a center-to-periphery discrepancy

Figure 3:

- To emphasize and quantify the change in parallelism, I would add measurement bars of the CCT and PCT and show how the ratio defers between normal and post-DMEK eyes</discuss></discuss></discuss></discuss></discuss>

7. PLOS authors have the option to publish the peer review history of their article (what does this mean?). If published, this will include your full peer review and any attached files.

Reviewer #2: No

---

## [Author Response · Author response to Decision Letter 3]

22 Aug 2020

Reviewer #2:

Thank you very much for your review of our work. Your comments have helped us to significantly improve our manuscript.



 Introduction

- Line 71: not sure what you mean by "because of injury". I would delete this. The sentence works without it: "However, this technique is associated with..."

Following your suggestion, we have deleted the word. (Line 71)

‘However, this technique is associated with disadvantages such as graft rejection, glaucoma (steroid-dependent) and suture-related problems, slow and low visual recovery with significant astigmatism, infection, or risk of globe rupture with trauma [2].’ (Line 71-74)

- Line 73: replace "heavy" with "significant"

Following your suggestion, we have revised the description. (Line 73)

‘However, this technique is associated with disadvantages such as graft rejection, glaucoma (steroid-dependent) and suture-related problems, slow and low visual recovery with significant astigmatism, infection, or risk of globe rupture with trauma [2].’ (Line 71-74)

- Line 74: replace "global rupture" with "risk of globe rupture with trauma"

Following your suggestion, we have revised the description. (Line 74)

‘However, this technique is associated with disadvantages such as graft rejection, glaucoma (steroid-dependent) and suture-related problems, slow and low visual recovery with significant astigmatism, infection, or risk of globe rupture with trauma [2].’ (Line 71-74)

- Line 81: need to define DSAEK acronym

Following your suggestion, we have added the abbreviation. (Line 81)

‘Previous studies have proved the superiority of DMEK to either Descemet’s stripping automated endothelial keratoplasty (DSAEK) or ultra-thin DSAEK in terms of visual function [12-16].’ (Line 81-83)

- Line 85: need space between "withcorneal"

Following your suggestion, we have revised the mistake. We are very grateful to you for the suggestion. (Line 86)

‘Previous reports have described factors associated with visual function and the specific changes in the posterior cornea after DMEK [12-16], with corneal backscatter and higher-order aberrations (HOAs) as main factors [12-20].’ (Line 85-87)

Materials and Methods

- Line 106: would change "ocular surface disease" to "ocular disease". I presume patients with glaucoma or retinal pathology were excluded due to effect on VA

Following your suggestion, we have revised the description. (Line 107)

‘Age-matched phakic eyes without history of ocular surgery or ocular disease were selected as healthy controls.’ (Line 106-107)

Results

- Line 160: need to define FECD acronym

Following your suggestion, we have revised the description. (Line 161)

‘Fuchs endothelial corneal dystrophy (FECD, n = 9)’ (Line 160-161)

Table 1

- Would use PBK acronym/definition instead of BK to keep consistent with text

Following your suggestion, we have revised the description to keep consistent with all the manuscript. (Table 1)

Table 2/3

- Would make text consistent between "Post-operative month 1" and "1 month after surgery"

- Could also consider shortening to Post-op month 1 or POM #1 for ease of reading the table, and then defining the acronym in the table legend/description

Following your suggestion, we have revised the description to the style of ‘Post-op month ~‘ with the acronym in the table legend. (Table 1)

Discussion

- Lines 268-327: these paragraphs have redundant/repeating elements - they all discuss discrepancy between central and peripheral endothelial cell density and how this may explain your results. The content of these paragraphs should be re-written so that each concept is discussed once and thoroughly.

"During DMEK, the central endothelium is stripped and replaced with an under-sized graft containing healthy endothelial cells. Post-operatively, these transplanted cells may migrate into the periphery to fill the area between the edge of the graft and the area of stripping, and come to a halt at the native peripheral endothelium due to contact inhibition. <>. This resultant discrepancy between high cell density in the center and low cell density in the periphery may account for the persistent changes between the post-DMEK eyes and healthy controls. As the central cornea deturgesces, the CCT normalizes but the PCT does not. <>. Since the deturgescence occurs mostly via the posterior stroma, the posterior KV improves while the anterior KV remains unchanged. <>. <> <>".

Following your suggestion, we have shortened the discussion. (Line 269-273)

Figure 2:

- Would modify (b) to more accurately reflect size of descemetorhexis and what you discuss in your text. A 9 mm descemetorhexis on an 11 mm cornea still leaves a peripheral population of native endothelial cells.

- Would maybe add an additional diagram showing how you hypothesize these centrally transplanted enothelial cells migrate to the periphery and result in a center-to-periphery discrepancy

Following your suggestion, we have already revised the Figure 2, and discussed our hypothesis in the discussion. 

Figure 3:

- To emphasize and quantify the change in parallelism, I would add measurement bars of the CCT and PCT and show how the ratio defers between normal and post-DMEK eyes

Following your suggestion, we have added the abbreviations and bars of CCT and PCT in Figure 3.

---

## [Decision Letter · Decision Letter 4]

28 Sep 2020

Optical characteristics after Descemet membrane endothelial keratoplasty: 1-year results

PONE-D-20-11108R4

Dear Dr. Hayashi,

We’re pleased to inform you that your manuscript has been judged scientifically suitable for publication and will be formally accepted for publication once it meets all outstanding technical requirements.

Kind regards,

Yu-Chi Liu, M.D

Academic Editor

PLOS ONE

Additional Editor Comments (optional):

Reviewers' comments:

Reviewer's Responses to Questions

**Comments to the Author**

1. If the authors have adequately addressed your comments raised in a previous round of review and you feel that this manuscript is now acceptable for publication, you may indicate that here to bypass the “Comments to the Author” section, enter your conflict of interest statement in the “Confidential to Editor” section, and submit your "Accept" recommendation.

Reviewer #2: All comments have been addressed

2. Is the manuscript technically sound, and do the data support the conclusions?

Reviewer #2: Yes

3. Has the statistical analysis been performed appropriately and rigorously? 

Reviewer #2: Yes

4. Have the authors made all data underlying the findings in their manuscript fully available?

Reviewer #2: Yes

5. Is the manuscript presented in an intelligible fashion and written in standard English?

Reviewer #2: Yes

6. Review Comments to the Author

Reviewer #2: Thank you to the authors for addressing all of my comments and suggestions. I hope I did not try your patience too much through 4 cycles of revisions! The final manuscript is much more polished and cohesive.

This is my last and final edit!

Introduction

- Line 81: DSAEK acronym expansion should go here (not line 83)

7. PLOS authors have the option to publish the peer review history of their article (what does this mean?). If published, this will include your full peer review and any attached files.

Reviewer #2: No

---

## [Editor Report · Acceptance letter]

30 Sep 2020

PONE-D-20-11108R4 

Optical characteristics after Descemet membrane endothelial keratoplasty: 1-year results 

Dear Dr. Hayashi:

I'm pleased to inform you that your manuscript has been deemed suitable for publication in PLOS ONE. Congratulations! Your manuscript is now with our production department. 

Kind regards, 

on behalf of

Dr. Yu-Chi Liu 

Academic Editor

PLOS ONE